# The ExtremeX global climate model experiment: Investigating thermodynamic and dynamic processes contributing to weather and climate extremes

Kathrin Wehrli[1], Fei Luo[2,3], Mathias Hauser[1], Hideo Shiogama[4], Daisuke Tokuda[5], Hyungjun Kim[5,6,7], Dim Coumou[2,3], Wilhelm May[8], Philippe Le Sager[3], Frank Selten[3], Olivia Martius[9,10,11], Robert Vautard[12], and Sonia I. Seneviratne[1]

[1]Institute for Atmospheric and Climate Science, Department of Environmental Systems Science, ETH Zurich, Zurich, Switzerland
[2]Institute for Environmental Studies, VU University Amsterdam, Amsterdam, Netherlands
[3]Royal Netherlands Meteorological Institute (KNMI), De Bilt, Netherlands
[4]Center for Global Environmental Research, National Institute for Environmental Studies, Tsukuba, Japan
[5]Institute of Industrial Science, University of Tokyo, Tokyo, Japan
[6]Moon Soul Graduate School of Future Strategy, Korea Advanced Institute of Science and Technology, Daejeon, Korea
[7]Department of Civil and Environmental Engineering, Korea Advanced Institute of Science and Technology, Daejeon, Korea
[8]Centre for Environmental and Climate Science (CEC), Lund University, Lund, Sweden
[9]Oeschger Centre for Climate Change Research, University of Bern, Bern, Switzerland
[10]Institute of Geography, University of Bern, Bern, Switzerland
[11]Mobiliar Lab for Natural Risks, University of Bern, Bern, Switzerland
[12]Institut Pierre Simon Laplace, Laboratoire des Sciences du Climat et de l'Environnement (LSCE), Gif sur Yvette, France

**Correspondence:** Kathrin Wehrli (kathrin.wehrli@env.ethz.ch)

**Abstract.** The mechanisms leading to the occurrence of extreme weather and climate events are varied and complex. They generally encompass a combination of dynamic and thermodynamic processes, as well as drivers external to the climate system, such as anthropogenic greenhouse gas emissions and land-use change. Here we present the ExtremeX multi-model intercomparison experiment, which was designed to investigate the contribution of dynamic and thermodynamic processes to recent

5   weather and climate extremes. The numerical experiments are performed with three Earth System Models: CESM, MIROC, and EC-Earth. They include control experiments with interactive atmosphere and land surface conditions, and experiments where either the atmospheric circulation, soil moisture or both are constrained using observation-based data. The temporal evolution and magnitude of temperature anomalies during heatwaves is well represented in the experiments with constrained atmosphere. However, the magnitude of mean climatological biases in temperature and precipitation are not greatly reduced

10  in any of the constrained experiments due to persistent or newly introduced biases. This highlights the importance of error compensations and tuning in the standard model versions. To show one possible application, ExtremeX is used to identify the main drivers of heatwaves and warm spells. The results reveal that both atmospheric circulation patterns and soil moisture conditions substantially contribute to the occurrence of these events. Soil moisture effects are particularly important in the tropics, the monsoon areas and the Great Plains of the United States, whereas atmospheric circulation effects are major drivers

15  in other mid- and high-latitude regions.

# 1 Introduction

Weather and climate extremes strongly affect society, human health, and ecosystems; therefore they need to be accurately simulated in numerical weather predictions and climate projections (e.g. Seneviratne et al., 2012). However, substantial biases remain in their representation in weather and climate models (e.g. Angélil et al., 2016; Maraun et al., 2017; Merz et al., 2020; Moon et al., 2018; Wehrli et al., 2018). For climate models used in the fifth phase of the Coupled Model Intercomparison Project (CMIP5), consistent biases can be found across models in the mean climatology of the lower atmosphere and land surface, for example for temperature and precipitation (Flato et al., 2013; Mueller and Seneviratne, 2014). These biases originate to some extent from the representation of the underlying processes driving evapotranspiration at the land surface (Mueller and Seneviratne, 2014), extratropical cyclones (Zappa et al., 2013), or the simulated sea ice and sea surface temperatures (SSTs) (Turner et al., 2013; Wang et al., 2014a). The difficulties in representing the mean climatology translate to biases in the representation of extreme events and impede their projection into the future. Therefore, an important question in the investigation of changes in extremes in a warming climate is the identification of the respective contribution of thermodynamic (thermal structure, water vapor and precipitation, land-atmosphere interactions) and dynamic (large-scale circulation) processes to their changes in occurrence and intensity (e.g. Pfahl et al., 2017; Shepherd, 2014; Trenberth et al., 2015; Wehrli et al., 2018, 2019; Zappa et al., 2015). Better isolating these contributions would help inform further model development as well as research on the attribution and projection of changes in weather and climate extremes (Vautard et al., 2016).

In the past, the number of record-breaking hot extremes has increased and it is expected to increase further if anthropogenic emissions continue to rise (e.g. Rahmstorf and Coumou, 2011; Shiogama et al., 2016; Power and Delage, 2019). Changes in the frequency, intensity and duration of various types of extremes can be seen in different regions of the world (e.g. Seneviratne et al., 2012). The most extreme events show the highest sensitivity to climate change (Seneviratne et al., 2014; Sillmann et al., 2013) and new, not yet seen extreme intensities are anticipated. These changes are related to a number of physical processes, their interactions with each other and their response to climate change.

The processes driving a specific extreme event and their relative importance can be studied in observation-based studies using linear regression (e.g. Arblaster et al., 2014; Wang et al., 2016; Dirmeyer et al., 2021) or forecast sensitivity experiments (e.g. Hope et al., 2016; Petch et al., 2020). In climate model simulations the role of the drivers can be studied by constraining the processes in the ocean, the atmosphere or at the land surface, which enables the study of drivers in isolation (e.g. Fischer et al., 2007; Hauser et al., 2016; Jaeger and Seneviratne, 2011). Recent work using these methods has shown that both soil moisture and atmospheric circulation play an important role in driving heatwaves (e.g. Dirmeyer et al., 2021; Petch et al., 2020; Suarez-Gutierrez et al., 2020). In this study, we present the new "ExtremeX" multi-model experiment in which, among other possible applications, the contribution of thermodynamic and dynamic processes to recent extreme events can be investigated in three Earth System Models (ESMs). The models used in ExtremeX are the Community Earth System Model version 1.2 (CESM1.2), the Model for Interdisciplinary Research on Climate version 5 (MIROC5) and the European Community Earth System Model version 3 (EC-Earth3). The purpose of this study is (1) to introduce the ExtremeX experiments, and (2) to

apply the introduced framework to study the drivers of heatwaves and to identify globally for which locations warm spells are generally dominated by processes at the land surface or by atmospheric circulation.

The ExtremeX experiment builds on the study of Wehrli et al. (2019). This previous study introduced a framework to disentangle the role of the ocean, atmospheric circulation, soil moisture conditions and recent climate change on extreme events. Therefore, experiments were carried out with prescribed SSTs and sea ice and additionally constraining the land surface and/or the atmosphere using observation-based conditions. Atmospheric variability was constrained using a grid-point nudging approach (Jeuken et al., 1996) to relax the horizontal winds toward reanalysis. This nudging approach has been verified for CESM1.2 (Wehrli et al., 2018) also analyzing biases for the nudged vs. non-nudged model climatologies, showing only minor changes to total biases. In Wehrli et al. (2019), soil moisture in the upper soil layers was constrained to control the role of land surface feedback on extreme events. The experiments with atmospheric nudging and/or prescribed soil moisture were then used to study recent heatwaves highlighting the combined role of dynamics (i.e. atmospheric circulation) and thermodynamics (in that case referring to land-atmosphere interactions) in driving these events.

Here, building further upon the studies with CESM1.2, we present the results of the same experiments but carried out for three ESMs that were contributed each by one of the collaborating modeling groups. The models do not show high interdependence and thus are an optimal selection for a small ensemble (Brunner et al., 2020; Knutti et al., 2013). EC-Earth3 is the most recent of the three models and, hence, has the highest horizontal and vertical resolution of the three models used here. Since EC-Earth3 was used for CMIP6, it also has the most recent forcing data. Among CMIP5 and CMIP6 models, EC-Earth3 is one of the best-performing models with regard to the representation of atmospheric circulation in the Northern Hemisphere mid-to-high latitudes (Brands, 2022; Fernandez-Granja et al., 2021). MIROC5 has contributed to CMIP5 as well as the 1.5°C versus 2.0°C global warming experiments (e.g. Hirsch et al., 2018; Mitchell et al., 2017; Shiogama et al., 2019).

The presented work expands on previous work in Wehrli et al. (2018, 2019) by quantifying biases of the near-surface climatology for different constraining experiments and three models. Near-surface temperature anomalies during four heatwaves evaluated in Wehrli et al. (2019) are compared in the three ESMs of ExtremeX. Additionally, warm spells are examined grid-point wise to identify the role of the atmosphere and soil moisture for different spell lengths. The research questions asked in the following are:

- Are model deficiencies in atmospheric circulation and the land surface contributing to climatological model biases? Are model biases reduced when these processes are constrained?

- Are the ExtremeX ESMs able to reproduce temperature anomalies of past heatwaves when constrained with observation-based data?

- What is the role of the physical climate drivers and climate change for four observed heatwaves?

- What is the relative contribution of the land surface and the atmospheric circulation to warm spells globally and how do the contributions vary regionally?

The ExtremeX experiments could also be used to examine other types of events than heatwaves. They are suitable for more in-depth analysis of model biases by examining for example the atmospheric moisture and heat budgets or the surface energy balance. In Luo et al. (2021) the ExtremeX experiments are used to study the origin of model biases in the anomaly of upper-level winds and near-surface climatology during certain summertime Rossby wave events in the Northern Hemisphere by constructing composites. Other applications have not been tested so far and will be left to explore in future studies.

In Sect. 2, we introduce the experimental design and methods used, in Sect. 3 we describe the three models composing ExtremeX and in Sect. 4 we evaluate the constraining of the atmospheric circulation and land surface. Then the framework is applied to investigate the contribution of atmospheric circulation patterns vs. soil moisture conditions for selected heatwaves between 2010 to 2015 (Sect. 5.1) and for the occurrence of warm spells (Sect. 5.2). The conclusions and outlook is given in Sect. 6.

## 2   Design of the model intercomparison project

The ExtremeX experiment was conducted in a collaboration of three modeling groups running three ESMs, whereof two (CESM and MIROC) were run in the CMIP5 configuration and one (EC-Earth) in the CMIP6 configuration. Five experiments were designed to unravel the source of model biases and to separate the contribution of atmospheric circulation and land surface conditions to extreme events. This is done by constraining the ocean, land surface and atmosphere using observation-based data. The experiments and methods used are described in the following, and overall follow the setup of Wehrli et al. (2019).

### 2.1   Experimental design

Five experiments are included in ExtremeX. All experiments prescribe SSTs, sea ice cover fraction, and land use (i.e. vegetation) but differ in the simulation of the atmospheric circulation and soil moisture that are either interactive or constrained. See Table 1 for an overview of the experiments. The control experiment (AI_SI, "Atmosphere Interactive, Soil Interactive") uses the standard setup where the atmosphere and soil moisture are both interactive. In the soil moisture experiment (AI_SF, "Atmosphere Interactive, Soil Forced") the atmosphere is interactive and soil moisture is constrained, and vice versa for the nudging experiment (AF_SI, "Atmosphere Forced, Soil Interactive"), where the latter is only used to validate the atmospheric nudging in this study. Finally, in the fully constrained (AF_SF, "Atmosphere Forced, Soil Forced") and soil moisture climatology (AF_SC, "Atmosphere Forced, Soil Climatological") experiments both components are constrained prescribing soil moisture varying over time or prescribing soil moisture using climatological soil moisture (but including the seasonal cycle), respectively.

For each experiment, one or five simulations are initialized in 1979. The ensembles for AI_SI and AI_SF are enlarged from 5 to 100 members for 2009–2015/2016. The number of simulation runs for the other experiments (AF_SI, AF_SC and AF_SF) is constant. The small ensembles of five members for AF_SI from MIROC and CESM were used to confirm that variability between the members is highly reduced for winds at the surface by nudging the higher model levels, even though in the

**Table 1.** Overview of experiments and configuration of the model components. The number of simulation runs for the experiments is given for the two successive simulation periods (# 1979–2008 and # 2009-2015/2016) and for the models in the order: CESM1.2, EC-Earth3, MIROC5.

| Name | Acronym | Atmosphere | Soil Moisture | Ocean | # 1979–2008 | # 2009–2015/2016 |
|------|---------|------------|---------------|-------|-------------|------------------|
| Control | AI_SI | interactive | interactive | prescribed | 5, 5, 5 | 100, 100, 100 |
| Soil moisture experiment | AI_SF | interactive | prescribed | prescribed | 5, 5, 5 | 100, 100, 100 |
| Nudging experiment | AF_SI | nudged | interactive | prescribed | 5, 1, 5 | 5, 1, 5 |
| Fully constrained | AF_SF | nudged | prescribed | prescribed | 1, 1, 5 | 1, 1, 5 |
| Soil moisture climatology | AF_SC | nudged | prescribed* | prescribed | 1, 1, 5 | 1, 1, 5 |

* Soil moisture is prescribed to the 1982–2008 climatology of the reconstructed soil moisture time series.

ExtremeX setup, winds are unconstrained in the lowest model levels (Sect. 2.2.1, see also Sect. 4 and Fig. A1). The analysis of the simulations starts in 1982 because the first three simulation years are regarded as spin-up. Likewise, 2009 is regarded
as the spin-up for the additional members to diverge and therefore analysis of the years 2010–2015/2016 is recommended for example for event-based analysis of extremes, as is done here.

For the simulations, natural and anthropogenic forcing was used from observation-based data during the observational period and from scenarios thereafter (Sect. 3). The three models choose different scenarios, either the Representative Concentration Pathway 8.5 (RCP8.5; van Vuuren et al., 2011) (CESM), RCP4.5 (MIROC) or the Shared Socio-economic Pathway 3-7.0
(SSP3-7.0; Meinshausen et al., 2020) (EC-Earth). The choice between RCP4.5, RCP8.5 and SSP3-7.0 is not expected to affect the results in this study as the scenarios barely differ for the considered time period.

## 2.2 Methods

The ocean, land surface and atmosphere are constrained to follow observation-based data, either time-varying or climatological
ones. All experiments are conducted with SSTs and sea ice concentration prescribed. In the following, the methods to constrain the atmospheric circulation and the land surface are described. We use the term "constraining" to generally refer to the applied method of nudging the atmospheric large-scale circulation and prescribing soil moisture.

### 2.2.1 Atmospheric circulation nudging

Nudging the atmospheric circulation in a climate model strongly reduces the dynamic variability in a simulation. For ExtremeX,
all modeling groups use a grid-point nudging approach (Jeuken et al., 1996) to constrain the atmospheric large-scale circulation by adding a tendency term to the prognostic equations of the zonal and the meridional winds:

$$\frac{\partial U}{\partial t} = ... - \frac{K(z)}{\tau}\Big(U(x,t) - U_{target}(x,t)\Big) \tag{1}$$

The term to the right hand side of Equation 1 is computed from the difference between a reference data set $U_{target}$ and the computed model value ($U$). It is weighted by a relaxation time scale $\tau$ (Kooperman et al., 2012), which controls the strength of the constraint. A very short relaxation time scale means a strong constraint of the dynamics while a long relaxation time scale allows larger deviations from the reference. The length of the relaxation time scale is chosen such that it does not dominate model physics but guarantees good agreement with the reference data set. All three models use a 6 h relaxation time scale following Kooperman et al. (2012). The reference data is given by 6-hourly wind fields from the ERA-Interim reanalysis (Dee et al., 2011), which are linearly interpolated to the model time step and regridded to the model resolution. At each model time step, the nudging term is used to update the horizontal wind variables. Additionally, a height-dependent weighting $K(z)$ is introduced, enabling a free evolution of the boundary layer while the nudging strength increases with height and controls the large-scale circulation (mostly above 700 hPa). The exact implementation of the height-dependent profile was chosen by the groups individually to fit their respective models (Fig. 1).

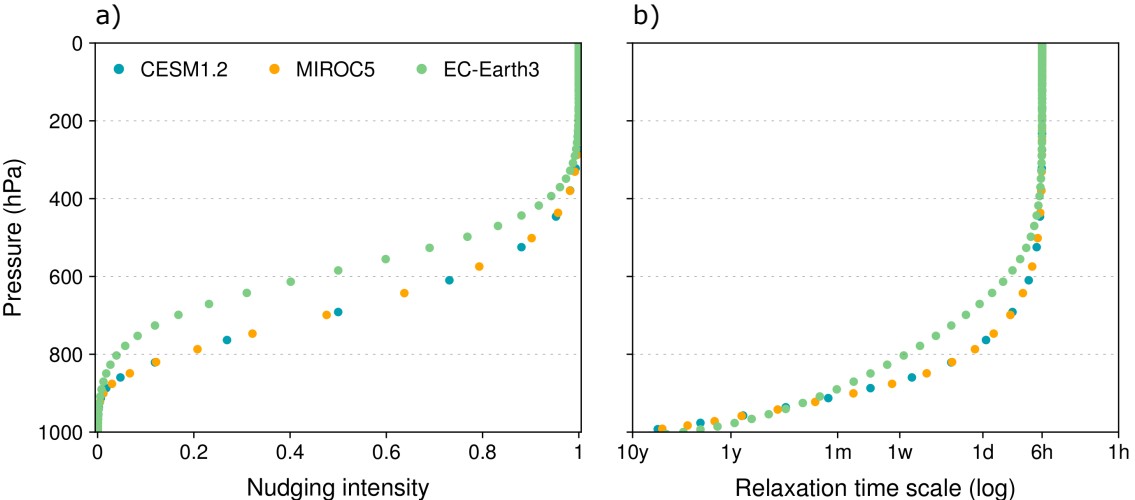

**Figure 1.** Nudging profile for the three ExtremeX ESMs. The height-dependent nudging intensity ($K(z)$, a) and resulting relaxation time scale from the division by $\tau$ (b) are marked with dots for each model. The nudging intensity is given from zero (no nudging) to one (fully nudged).

### 2.2.2 Prescribing soil moisture

Prescribing soil moisture in a climate model enables the isolation of the main effects of land surface conditions and feedback on climate. In an experiment with interactive atmosphere but prescribed soil moisture (AI_SF), the circulation will adapt to the given land surface conditions, but there is no feedback in the opposite direction. Hence, the land is decoupled from the atmosphere. In the present set-up, soil moisture is prescribed to reflect observed conditions, similar to the SST-driven and nudged-circulation simulations. However, directly prescribing soil moisture from observations or observations-based products

(e.g. reanalyses) in a climate model can lead to inconsistencies due to differences in model climatologies and soil parameterisations (Koster et al., 2009). Instead, soil moisture reconstructions are generated by driving the land surface module of the respective ESM with meteorological fields from reanalysis data (e.g. air temperature, humidity, wind, precipitation, radiation). The generated daily or 6-hourly soil moisture constitutes the model-specific soil moisture reconstruction. This allows soil moisture in the prescription experiments to follow observed-based soil moisture states, while still being in balance with the model

climatologies and land model parameterisations. The method to constrain the land model with soil moisture reconstructions is inspired by the approach developed by Hauser et al. (2017b) for CESM. Not all models followed the approach in every aspect, as it was adapted to the respective model and tools available (see details in the individual model descriptions in Sect. 3). The common idea is that the model hydrology is active, but at the end of each time step the modelled soil moisture is replaced with the target soil moisture from the reconstruction.

### 2.3 Reference data sets

The atmospheric nudging is validated using winds from ERA-Interim (Dee et al., 2011) as reference. Monthly near-surface temperature is retrieved at 0.5° resolution from the Climatic Research Unit, University of East Anglia (Harris et al., 2020), using version 4.03 of the data set (CRUTS in the following). Precipitation data at 0.5° resolution is obtained from the Global Precipitation Climatology Centre full data product version 2018 (GPCC-FD; Ziese et al., 2018) and at 0.1° resolution from the

Multi-Source Weighted-Ensemble Precipitation data set, Version 2.2 (MSWEP; Beck et al., 2018). As reference for evapotranspiration the long, merged synthesis product (based on all data set categories) from the LandFlux-Eval data set at 1° horizontal resolution is used (Mueller et al., 2013). Total cloud cover information was retrieved from the International Satellite Cloud Climatology Project D1 (ISCCP-D1; Rossow and Schiffer, 1999) at 2.5° resolution. All reference data sets are regridded to the original resolution of each model for the comparison.

### 2.4 Disentangling approach

In Sect. 5, we disentangle the contribution of the physical drivers (atmospheric circulation, land surface conditions and the ocean state) and of recent climate change to temperature extremes. The method is briefly explained below, readers can refer to Wehrli et al. (2019) for more details. The disentangling method only takes anomalies of a variable with respect to the experiment climatology into account. In this study, the disentangling is applied to anomalies of daily mean temperature and

daily maximum temperature (TX). The different contributions are assumed to be additive and, hence, differences between the experiments are computed as shown in Fig. 2.

The fully constrained experiment (AF_SF) is taken as the "model truth" because it is as close to observations as the model can get. Therefore, AF_SF is set to 100 % of the event and the disentangling method determines what fraction of the event anomaly is explained by the other experiments. First, the contribution of recent climate change is computed as the anomaly of

the years 2010–2015/2016 during the same time of the year the event took place (but excluding the event year) in AI_SI with respect to its 1982–2008 climatology. Note that a small fraction of the 2010–2015/2016 anomalies related to the prescribed SST conditions, could also be due to decadal variability. The anomaly of AI_SI at a specific point in time (e.g. during an extreme

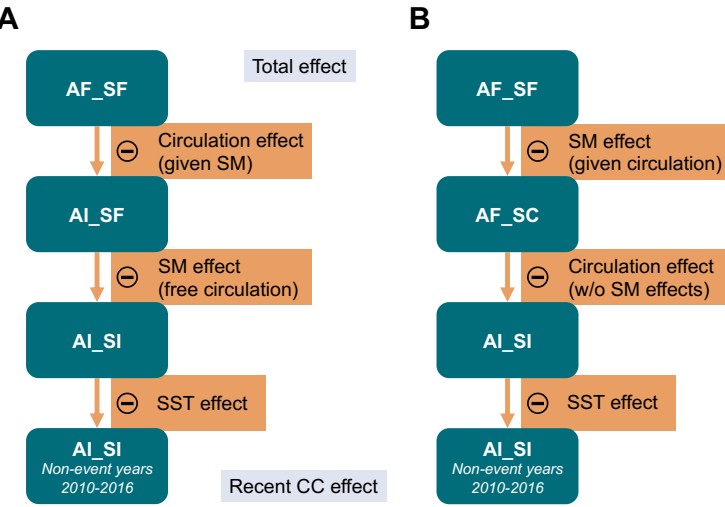

**Figure 2.** Schematic showing model experiments and effects isolated for temperature extreme events. For the experiments, the anomaly during the event (or during the same time of the year but for non-event years) is considered relative to the 1982–2008 climatology (turqouise rectangles). The magnitude of the anomaly in AF_SF is taken as the total effect and the anomaly of AI_SI during non-event years is taken as the recent climate change (CC) effect (grey boxes). Further effects are disentangled by computing differences between the experiments along the orange arrows as indicated by the minus sign (orange rectangles). Two approaches (black letters A and B) are followed differing in how soil moisture (SM) contributions are separated from atmospheric circulation contributions.

event) compared to its 1982–2008 climatology is a combination of recent climate change (i.e. warming since the climatology period, which is estimated in the first step using the anomaly of non-event years from AI_SI), the observed SST pattern, and natural variability. The natural variability is controlled by using a large ensemble of 100 members. Hence, following the additive assumption, the remaining anomaly of AI_SI corresponds to the contribution of the ocean. To estimate the contribution of the land surface state (i.e. soil moisture) and the atmospheric circulation, the disentangling method by Wehrli et al. (2019) follows two approaches. The first approach (A) is to quantify the contribution of soil moisture as the anomaly in AI_SF minus the anomaly in AI_SI, and the contribution of the atmospheric circulation as the anomaly in AF_SF minus the anomaly in AI_SF. The second approach (B) is to quantify the contribution of soil moisture as the anomaly of AF_SF minus the anomaly of AF_SC, and the contribution of the atmospheric circulation as the anomaly of AF_SC minus the anomaly of AI_SI. Wehrli et al. (2019) show that the two approaches give similar results, which is confirmed here (Sect. 5). Hence, in this study the results from approaches A and B are averaged, giving equal weight to both. The results for the single approaches are also documented in the Appendix. The individual data analysis carried out for four recent heatwaves in Sect. 5.1 and warm spells in Sect. 5.2 is described in the respective sections.

# 3 Model descriptions

## 3.1 Community Earth System Model (CESM)

The Community Earth System Model is run in Version 1.2 (CESM1.2; Hurrell et al., 2013). Coupled are the Community Atmosphere Model Version 5.3 (CAM5; Neale et al., 2012) and the Community Land Model Version 4 (CLM4; Lawrence et al., 2011; Oleson et al., 2010). Both are run on a horizontal resolution of $0.9° \times 1.25°$. The atmosphere model, CAM5, uses hybrid sigma-pressure coordinates and has 30 vertical layers. CLM4 has 15 soil layers, whereof active hydrology is computed in the upper 10 layers (down to 3.8 m).

Natural forcings as well as forcings from greenhouse gases (GHGs), aerosols and land-use change follow the setup in Wehrli et al. (2019). Major GHGs ($CO_2$, $N_2O$ and $CH_4$) are prescribed to observed global values whereas other anthropogenic forcings follow RCP8.5 after 2005. A merged product of the Hadley Centre sea ice and SST data set Version 1 (HadISST1) and the NOAA weekly optimum interpolation SST analysis Version 2 (OI2) was used to prescribe transient monthly observations of SSTs and sea ice concentrations (Hurrell et al., 2008). For prescribing soil moisture, the prescription method developed by Hauser et al. (2017b) was used, which replaces the model-calculated soil moisture value by a target value at the end of each model time step. The prescribed target soil moisture is computed by running CLM4 offline driven by reanalysis data from ERA-Interim. Below $0°C$ soil temperature in the model, soil moisture is computed interactively whereas at warmer temperatures the soil liquid water is prescribed to the total (liquid + ice) soil moisture of the target data set. This prevents artificial creation of ice, which can produce unrealistic heat fluxes (Hauser et al., 2017b).

## 3.2 Model for Interdisciplinary Research on Climate version 5 (MIROC5)

The Model for Interdisciplinary Research on Climate version 5 (MIROC5) was developed jointly at the Atmosphere and Ocean Research Institute, University of Tokyo; National Institute for Environmental Studies; and Japan Agency for Marine-Earth Science and Technology (Watanabe et al., 2010). The horizontal spectral resolution of the model is T85 (wave number 85 with triangular truncation corresponding to a horizontal resolution of about 160 km). There are 40 vertical model levels of hybrid sigma-pressure coordinates up to 3 hPa. The land scheme of MIROC5 is an updated version of Minimal Advanced Treatments of Surface Interaction and Runoff (MATSIRO; Takata et al., 2003), which predicts the temperature and water in six soil layers down to a depth of 14 m, one canopy layer, and three snow layers. The SST and sea ice concentration data of HadISST1 were used (Rayner et al., 2003). See Shiogama et al. (2013) for the setup of natural (solar irradiance and volcanic activity) and anthropogenic (GHGs; sulfate, black and organic carbon aerosols; ozone; land use land cover change) forcing agents. The anthropogenic forcing agents after 2005 were based on RCP4.5.

For prescribing soil moisture in MIROC5, the model replaces the calculated soil moisture with a target value at the beginning of each model time step. The prescribed target soil moisture was simulated by the land scheme in offline mode driven by atmospheric fields from ERA-Interim. To remove negative values of liquid soil moisture content, the replacing procedure also limits ice content so that the total soil moisture does not exceed the prescribed one.

### 3.3 European Community Earth System Model Version 3 (EC-Earth3)

The European Community Earth System Model (EC-Earth3) is a climate model with the atmosphere component based on the European Centre for Medium-Range Weather Forecasts' (ECMWF) Integrated Forecasting System (IFS) and it is maintained by the EC-Earth consortium. Version 3.3.1 with IFS cycle cy36r4 was used for the ExtremeX experiments. The horizontal spectral resolution of the model is T255 (about 80 km), and there are 91 vertical model levels. The model has a hybrid sigma/pressure coordinate system and a reduced Gaussian grid (N128). All experiments were produced with the SSP3-7.0 CMIP6 scenario and prescribed monthly ocean fields from the merged HadISST1 and NOAA OI2 data set with pre-applied SST/sea ice consistency checks (Hurrell et al., 2008). More information regarding the model can be obtained from www.ecmwf.int, www.ec-earth.org, and Döscher et al. (2022) for greenhouse gases, aerosols and land-use prescribed in EC-Earth3.

For prescribing soil moisture in EC-Earth, the simulated soil moisture is replaced by the respective target values for each of the four soil layers at the end of each model time step. As target values, 6-hourly soil moisture data from ERA-Interim/Land (Balsamo et al., 2015) were used. ERA-Interim/Land uses the H-TESSEL land surface model (Balsamo et al., 2009) which has four soil layers, covering 0 cm–7 cm, 7 cm–28 cm, 28 cm–100 cm and 100 cm–255 cm of the soil.

## 4 Validation of the constrained atmosphere and soil moisture experiments

In the setup used for this study, the atmospheric nudging is stronger in the upper atmosphere and close to zero at the surface (Fig. 1). Hence, it can be expected that the variability between ensemble members is strongly reduced, especially at higher atmospheric levels, and that the simulated winds closely follow the winds in ERA-Interim with increasing nudging strength. This is confirmed by evaluating the wind fields of the three models at the grid point level. For MIROC (and less visibly also for CESM; Fig. A1) there is some variability in wind speed and direction between the five members of the AF_SI ensemble at near-surface levels, whereas at 500 hPa and above all members are nearly identical and an almost exact representation of the reference wind speed and direction. For EC-Earth, only one simulation of AF_SI was run. Although the nudging profile of EC-Earth is shifted to higher altitudes compared to the other two models (Fig. 1), the horizontal winds represent the reference equally well for the selected pressure levels (Fig. A1; see also Fig. A2). For all models nudging the large-scale atmospheric flow also has a strong control on near-surface winds.

In the following, the climatological model biases are compared between the experiments. First, regional and global RMSEs are examined in Sect. 4.1. Then, the sign and location of seasonal biases is discussed in Sect. 4.2.

### 4.1 Global and regional biases in surface temperature and precipitation

Intuitively, one might expect that biases with respect to observations are reduced when either soil moisture or atmospheric circulation or both are constrained. Near-surface temperature, for example, is driven by radiative processes and surface turbulent fluxes. The incoming radiation is related to the abundance, thickness and composition of clouds, which is parameterized in the

models and driven by weather systems (e.g. Bony et al., 2015). Surface turbulent fluxes are driven by soil moisture availability,
which affects the partitioning in sensible and latent heat fluxes, especially in transitional climate regimes (e.g. Miralles et al.,
2019; Santanello et al., 2018; Seneviratne et al., 2010). Similarly, precipitation is affected by both soil moisture and atmospheric
circulation. The location and intensity of rainfall and snow is driven by the passage of low and high pressure systems as well
as by soil moisture conditions (e.g. van der Ent et al., 2010; Guillod et al., 2015; Moon et al., 2019).

In the following, we quantify the near-surface temperature and precipitation biases in the experiments. Therefore, the model
climatologies are compared to a reference by computing the root mean square errors (RMSEs) for the seasonal and annual
averages of the 1982–2008 climatology. The mean over all simulation runs was computed where multiple members were
available. Only land grid points are considered (except Antarctica).

In the global and annual average, the RMSEs in the experiments with nudged atmosphere and/or prescribed soil moisture
are nearly equal to the RMSE in AI_SI (Fig. 3). For EC-Earth the temperature bias increases when soil moisture is prescribed
in AI_SF and AF_SF (Fig. 3a). For MIROC a large precipitation bias is introduced when nudging the atmospheric circulation
in AF_SI (Fig. 3b). However, the bias is reduced in the fully constrained experiment (AF_SF). Temperature biases are largest
during the December-January-February (DJF) season in CESM and EC-Earth. In MIROC they are largest during the June-
July-August (JJA) season, except for AF_SF where biases are largest in DJF. Precipitation biases are largest in JJA for all
models. For the annual regional averages, large temperature biases can be found in regions with sparse observational coverage
such as GIC (Greenland/ Iceland), NEN (northeastern North America) and RAR (Russian Arctic; Iturbide et al., 2020, for an
overview of the AR6 reference regions see Fig. A3). Large temperature biases can also be found in small regions, which are
only represented by a few grid points in the models used, such as NWS (northwestern South America), SWS (southwestern
South America) and NZ (New Zealand). Additionally, the complex topography of the Himalayas (TIB) and Andes (SWS and
NWS) is also a likely source of temperature biases. Regional precipitation biases are generally larger in wet regions such as the
Amazonian regions SAM (South American Monsoon) and NSA (northern South America); and the regions in Central Africa,
namely CAF (central Africa), SEAF (southeastern Africa) and NEAF (northeastern Africa); as well as the monsoon regions
SEA (southeastern Asia) and SAS (southern Asia). Precipitation biases are also larger for small regions.

In general, the experiments with nudged atmosphere and/or prescribed soil moisture do not show a significant reduction of
the surface climatology RMSEs in any of the models or for any region of the world. In many cases, constraining the components
of the model leads to even larger biases. This contradicts the initial intuitive assumption and suggests that no sole component
of the model is responsible for the biases. Hence, the climatological biases that are discussed here cannot be corrected by
improving the representation of the model components in isolation.

## 4.2 Location and sign of seasonal biases

In the Northern Hemisphere mid-latitudes, the control simulations (AI_SI) for the CESM and MIROC models are systemat-
ically too hot (Fig. 4a) and in some regions also too dry (Fig. 5a) during boreal summer (JJA). North America, Central Asia
and Eastern Europe show the largest biases. This agrees with the findings by Wehrli et al. (2018) for CESM. For EC-Earth,
only Central Asia and parts of the Midwestern United States (U.S.) are too hot and dry, while other regions are mostly too

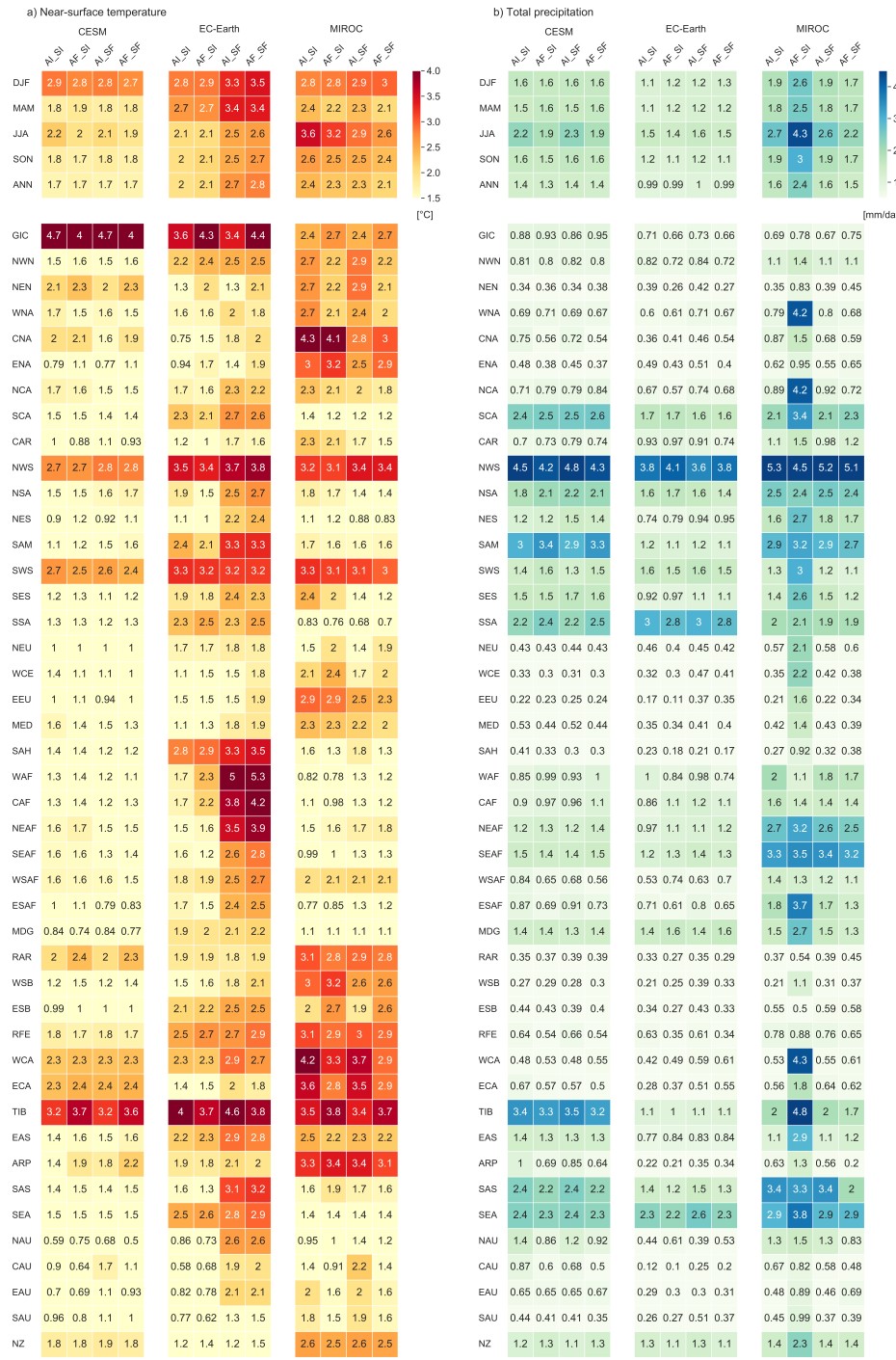

**Figure 3.** Root mean square errors (RMSEs) for a) near-surface temperature and b) total precipitation. The top section shows the global RMSE for land grid points and seasonal and annual (ANN) averages. The bottom section shows the average annual RMSE for the reference regions from the Sixth Assessment Report (AR6) on climate change (Iturbide et al., 2020). For an overview of the AR6 reference regions see Fig. A3. Ocean grid points and Antarctica are excluded.

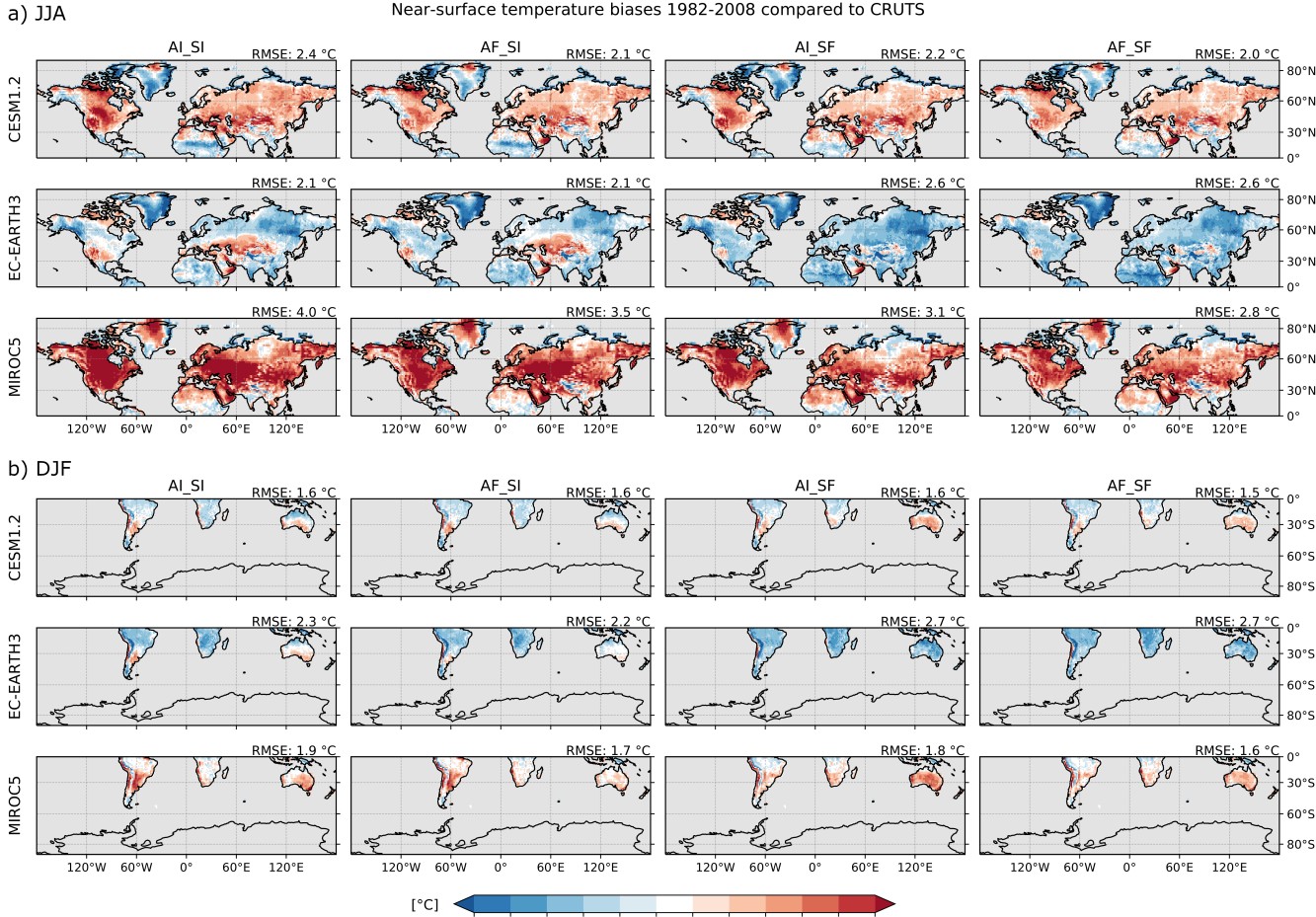

**Figure 4.** Bias in near-surface temperature (2m) with respect to CRUTS. Shown is the seasonal average over (a) June-July-August (JJA) for the Northern Hemisphere and (b) December-January-February (DJF) for the Southern Hemisphere. Ocean grid points and Antarctica are masked out in grey. The root mean square error (RMSE) averaged over all land grid points of the respective hemisphere is given in the upper right corner of each experiment and model.

cold and wet. In all models, the regions where JJA temperature is overestimated coincide with regions where cloud coverage is underestimated (Fig. A4). Especially in MIROC, a large negative cloud cover bias can be found for the Northern Hemisphere

295  midlatitudes. In MIROC, large areas in central and eastern North America, eastern Europe and Asia show a negative evapotranspiration bias as well (Fig. A5). In CESM and EC-Earth evapotranspiration is underestimated in central Asia and parts of western North America. This indicates that the warm temperature biases are related to underestimated evapotranspiration and cloud coverage in JJA.

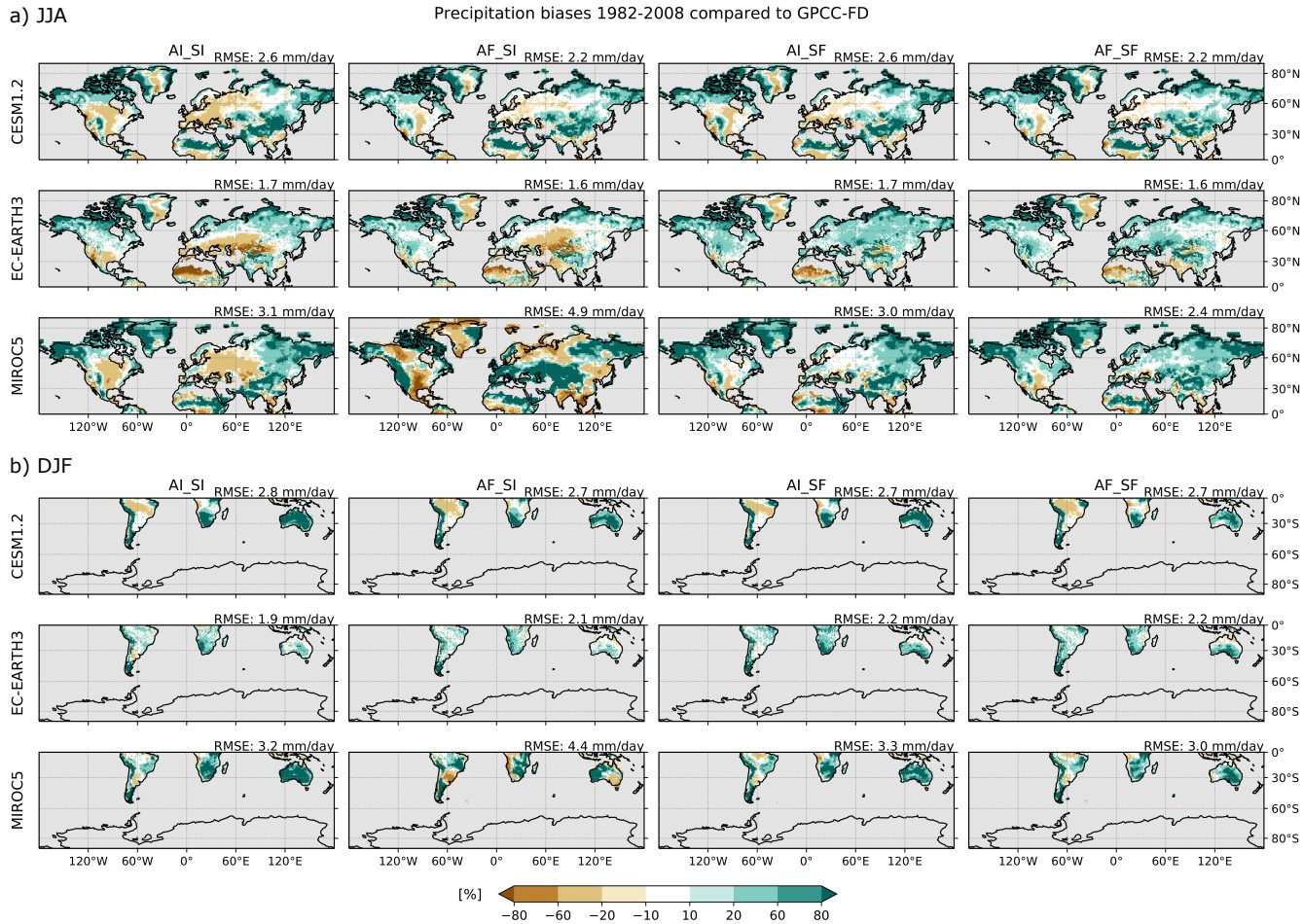

**Figure 5.** Bias in total precipitation with respect to GPCC-FD. Shown is the seasonal average over (a) JJA for the Northern Hemisphere and (b) DJF for the Southern Hemisphere. Values are plotted as percentage deviation from the reference data set. Ocean grid points, Antarctica, as well as grid points with a seasonal average of less than 0.1 mm precipitation per day in the reference data set are masked out in grey. The RMSE averaged over all valid grid points of the respective hemisphere is given in mm per day in the upper right corner of each experiment and model.

Nudging the atmosphere in AF_SI reduces some of the JJA temperature biases in the Northern Hemisphere in CESM and MIROC (Fig. 4a). For MIROC, large precipitation biases are introduced with nudging (Fig. 5a). Some of the midlatitude regions change from too little precipitation to too much and vice versa. Hence, correcting the atmospheric circulation seems to lead to an overcompensation of biases in MIROC. In EC-Earth, nudging does not strongly affect the JJA temperature and precipitation climatology. Constraining the soil moisture in AI_SF leads to a reduction of the hot and dry bias in the Northern Hemisphere midlatitudes in MIROC. For CESM, the changes are smaller but there is a reduction of the hot and dry bias in Europe and the U.S. Midwest. For EC-Earth, constraining the soil moisture, however, introduces or increases the cold and dry bias nearly everywhere. The fully constrained experiment (AF_SF) is the experiment with the smallest temperature and precipitation biases for CESM and MIROC, suggesting that at least for these models, a correct representation of atmospheric circulation patterns and soil moisture conditions can improve the models' overall performance. Nonetheless, for EC-Earth AF_SF shows larger climatological temperature biases than AI_SI (Fig. 4), and for precipitation, the biases remain of similar magnitude (Fig. 5). This indicates that even if some part of the temperature and precipitation biases is reduced by constraining the atmospheric circulation and soil moisture in the model using observation-based data, other biases can be enhanced or change sign, resulting in a worse overall performance.

The results for the austral summer (DJF) confirm the findings for the Northern Hemisphere in JJA. For EC-Earth, AI_SF introduces and increases a cold (Fig. 4b) and wet bias (Fig. 5b) in the entire Southern Hemisphere. MIROC shows large precipitation biases for AF_SI, which are in certain places of the opposite sign but similar or larger magnitude than in AI_SI. For CESM there are only very small differences between the experiments.

All models show substantial biases in total cloud cover fraction (Fig. A4) and evapotranspiration (Fig. A5), which match the biases found in temperature (e.g. negative cloud cover bias and negative evapotranspiration bias for areas that are too warm in the model) and sometimes also for precipitation. Both variables rely heavily on parameterisations. An alternative explanation for why biases are still prevalent after correcting the atmospheric flow and the land surface is that ESMs are tuned to match, e.g., the radiation balance at the top of the atmosphere and global mean values of variables like near-surface temperature, clouds or sea ice (Mauritsen et al., 2012). When single components of the models are constrained using more realistic fields from observational products, the model components are no longer in balance with each other. This can result in an overcompensation of biases, as can be seen for example for precipitation in MIROC (Fig. 5). It is known that MIROC5 shows biases in the North Atlantic storm track activity compared to ERA-Interim (e.g. Brands, 2022; Zappa et al., 2013). Correcting this circulation bias in AF_SI leads to even larger precipitation biases, which are only reduced when the soil moisture is constrained as well.

The seasonal precipitiation climatology from the models was also compared to MSWEP (Fig. A6), confirming the above findings. The results for DJF for the Southern Hemisphere are very similar to the biases using GPCC-FD as reference. For JJA for the Northern Hemisphere, the models are more on the dry side when compared to MSWEP than if GPCC-FD is used as reference. The hemisphere-averaged RMSEs are in both cases very similar. Overall, the biases are not substantially reduced in any of the models when nudging the atmospheric circulation and/or prescribing soil moisture using observation-driven reconstructions. This shows that model biases are not primarily caused by the misrepresentation of large-scale atmospheric motion or soil moisture conditions. Instead, the biases might be caused by other processes such as radiation and cloud processes,

convection, precipitation, as well as land surface properties (e.g. land cover and land use, topography) and processes unresolved due to the model grid scale such as mesoscale circulations and sub-grid surface heterogeneity. The results also suggest that the models are tuned to have low temperature and precipitation biases in the interactive setup (with prescribed ocean).

## 5  Disentangling the contribution of physical drivers and climate change to recent heatwaves

In the previous section it was shown that large biases remain in the model climatology even if observation-based conditions are used to constrain the models. Nevertheless, nudging the large-scale atmospheric circulation and prescribing the soil mois­ture results in simulations that can accurately reproduce the temporal evolution and relative magnitude of events. This was shown by Wehrli et al. (2019) using CESM for five recent heatwaves, considering anomalies of TX. Hence, the presented set of experiments can be used to analyse extreme events if anomalies are used or a more elaborate bias-correction method is performed (e.g. Wehrli et al., 2020). In Sect. 5.1, four recent heatwaves are examined and it is shown that all three models accurately reproduce TX anomalies during and prior to heatwaves, when constrained with observation-based data. Then, the contribution of physical drivers and climate change is disentangled for the four heatwaves. The events chosen are: the 2010 Russian heatwave, the 2015 European heatwave, the 2012 heatwave in the U.S. (also known as the Midwest heatwave), and the Australian heatwave of 2012/2013. All events had drastic consequences to the local communities and the economies due to e.g. damages to agriculture, wildfires and increased mortality. They were investigated in numerous previous studies including Wehrli et al. (2019). The events were chosen due to their severity and impact as well as to ensure consistency and comparability with Wehrli et al. (2019). In Sect. 5.2 warm spells (during the warm season) are analysed grid point-wise to identify the relative contribution of atmospheric circulation vs. soil moisture.

### 5.1  Driving processes of recent heatwaves

For four heatwaves, the relative contributions of atmospheric circulation, soil moisture, ocean conditions and climate change (since 1982–2008) to TX anomalies are determined. As in Wehrli et al. (2019), spatial averages are taken over the event region and daily mean near-surface temperatures from ERA-Interim are used to identify the events. The hottest 15-day period defines the event and TX during this period is examined. Ocean grid points are excluded from the analysis.

TX anomalies (with respect to 1982–2008) for the heatwaves and previous months are shown in Fig. 6. Overall, the fully con­strained experiments (AF_SF) from all models agree well with temperature anomalies from reanalysis and among each other. Small deviations are found during the 2010 Russian heatwave for MIROC, which underestimates the temperature anomaly (Fig. 6a), and during the 2012 U.S. heatwave where CESM overestimates the temperature anomaly (Fig. 6c). TX anomalies for the same events from the nudging experiment (AF_SI) already compare well to ERA-Interim (Fig. A7) capturing the temporal evolution of TX anomalies similarly well as AF_SF. Correlation of near surface temperature anomalies between the experi­ments with atmospheric nudging and with ERA-Interim is very high. This confirms that observed surface anomalies can be accurately reproduced when nudging the atmospheric circulation. For MIROC and CESM, which both have five AF_SI simula-

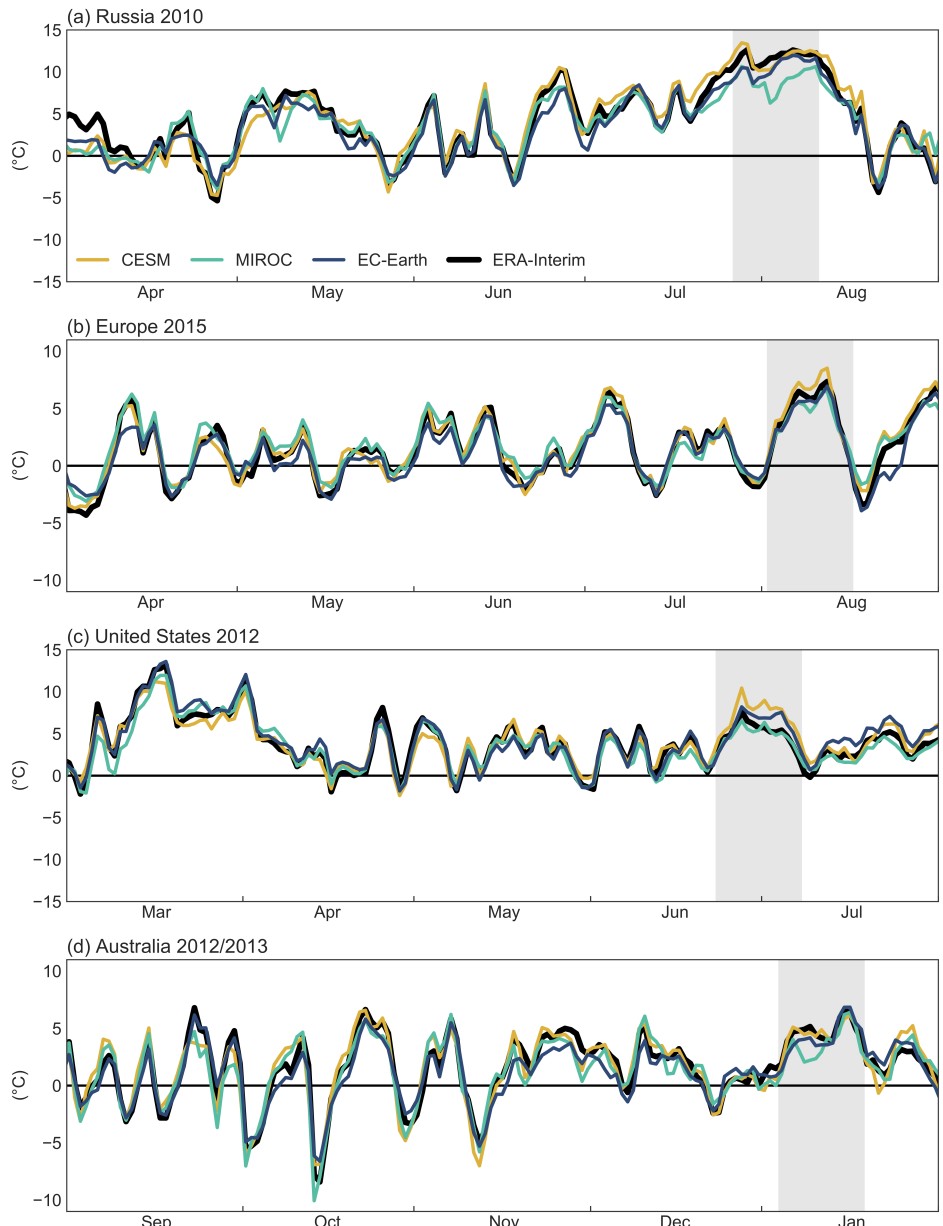

**Figure 6.** Daily maximum temperature anomaly compared to the 1982–2008 climatology for the fully constrained (AF_SF) experiment for the three models and for ERA-Interim (black line). The 15-day event period is highlighted in light grey.

tion runs, it can also be seen that nudging the atmosphere strongly constrains variability between ensemble members (Fig. A7). In the following, the four heatwaves are analysed separately.

The Russian heatwave of 2010 was characterized by extremely high temperatures over a long time period from late June to mid-August. A persistent blocking anticyclone was associated with the heatwave (e.g. Barriopedro et al., 2011; Trenberth and Fasullo, 2012). Due to early snow melt in the year and a deficit of precipitation, water scarcity was exacerbating the heatwave
(Barriopedro et al., 2011). For the analysis of the Russian heatwave of 2010, regional averages are computed over 50°N to 60°N and 35°E to 55°E (see Fig. A3 for the region outline). The hottest 15-day period lasts from 27 July to 10 August 2010 (Fig. 6a). TX anomalies from ERA-Interim exceed 10°C during the event, which is captured well by CESM and EC-Earth. In MIROC the anomaly is somewhat weaker. In general, the three models agree on the contributions of the drivers to the TX anomaly (Fig. 7a). They estimate that recent climate change explains around 7%–10% of the event anomaly. CESM is the only
model which shows a negative ocean contribution of around −7%, whereas the role of the ocean is negligible in the other models. This result is supported by the studies by Dole et al. (2011) using initialized forecasts and Hauser et al. (2016) using an ESM, which both found a weak role of the ocean in explaining the Russian heatwave of 2010. In contrast, observation-based studies like Martius et al. (2012) and Trenberth and Fasullo (2012) linked the driving atmospheric circulation conditions to SST anomalies identifying the ocean as an important driver. In all three models the event is mostly driven by atmospheric
circulation and soil moisture, which agrees with existing literature. The ratio of the circulation contribution to the soil moisture contribution is 70:30 for EC-Earth and CESM and 80:20 for MIROC. Assessing the two approaches to disentangle atmospheric circulation from soil moisture contributions separately gives very similar results (Fig. A8a).

The European heatwave of 2015 consisted of four hot spells that were intensified by drought conditions through land-atmosphere feedbacks (Dong et al., 2016; Hauser et al., 2017a; Orth et al., 2016) The event is analysed over the Western
and Central Europe (WCE) AR6 reference region (Iturbide et al., 2020, see Fig. A3; same as Central Europe (CEU) from previous assessment reports). The hottest 15-day period is from 2 August to 16 August 2015 (Fig. 6b). The magnitude of the TX anomaly before and during the heatwave is represented well by all models. However, there are differences in the attribution of the drivers. In EC-Earth, climate change contributes around 12% to the event anomaly, whereas in CESM climate change is estimated to contribute 22% and in MIROC 34% (Fig. 7b). EC-Earth and CESM agree that there is a small negative contribution
by the ocean of −9% and −7%, respectively. In MIROC the ocean is negligible for the event anomaly (around −1%). This is in contrast to the modeling study by Dong et al. (2016) and the observation-based study by Duchez et al. (2016) finding that the SST patterns set important preconditions for the 2015 European heatwave. The three ExtremeX models agree on the magnitude of the atmospheric circulation contribution, which is around half of the total event anomaly. However, the role of soil moisture depends on how much of the event anomaly is attributed to recent climate change. EC-Earth estimates the highest
relative soil moisture contribution with a ratio of 60:40 between circulation and soil moisture. The ratio is 70:30 for CESM and 75:25 for MIROC. The results for the two disentangling approaches differ most notably for EC-Earth, where the ratios are 65:35 for one approach (A) but nearly balanced for the other (B; Fig A8b).

The U.S. heatwave 2012 evolved concurrently with a severe drought after an unusually warm winter and spring (Dole et al., 2014; Hoerling et al., 2014; Wang et al., 2014b). The event is assessed for the region from 35°N to 50°N and 55°W to 110°W
(see Fig. A3) and for 23 June to 7 July (Fig. 6c). The TX anomaly is represented well by MIROC and EC-Earth and a bit overestimated in CESM. The models agree well on the relative contribution of the drivers. In EC-Earth and CESM, recent

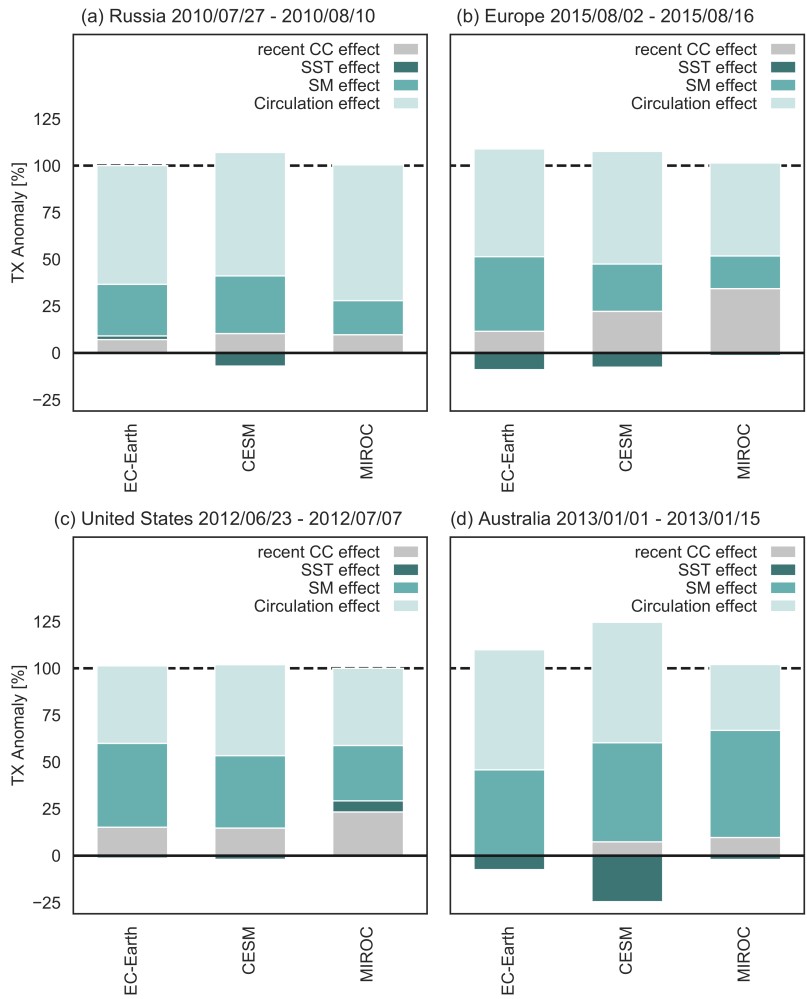

**Figure 7.** Contribution of recent climate change (since the 1982-2008 base period) and physical drivers to the daily maximum temperature (TX) anomaly during 4 recent events: a) Russia 2010, b) Europe 2015, c) United States 2012, d) Australian summer 2012/2013. The dates of the hottest consecutive 15-day period are given in the label of each subplot. The contributions of the drivers are normalized by the climatology of AF_SF for each model, respectively. The two approaches to compute SM vs. ATM contribution are merged giving equal weight to both.

climate change explains around 15% of the event anomaly, whereas for MIROC it is slightly more (23%, Fig. 7c). All models agree that the role of the ocean is very small, even if the sign is negative for EC-Earth and CESM (both $-1\%$) but positive for MIROC (6%). This agrees with Wang et al. (2014b) and Hoerling et al. (2014) who find the contribution by SSTs to be small.

The three models agree that the role of soil moisture conditions is about equal to the role of atmospheric circulation. This is supported by earlier studies finding an important contribution by both the weather patterns and soil moisture deficit (Hoerling et al., 2014; PaiMazumder and Done, 2016; Wang et al., 2014b). The ratio of circulation to soil moisture contribution is 50:50 for EC-Earth, 55:45 for CESM and 60:40 for MIROC. The individual results for the disentangling approaches show that for all models soil moisture dominates for one approach (A) while atmospheric circulation dominates for the other (B; Fig. A8c).

At its time, the summer of 2012/2013 was the warmest summer observed in Australia but it has since then been surpassed by the 2018/2019 and the 2019/2020 summers (Bureau of Meteorology, 2020). The Australian heatwave of 2012/2013 is analysed for the region from 18°S to 30°S and 133°E to 147°E (see Fig. A3). The hottest consecutive 15-days occur just at the beginning of 2013 from 1 January to 15 January 2013 (Fig. 6d). The models represent the TX anomaly from ERA-Interim mostly well, except that in MIROC it is underestimated during the first half of the event period. While the contribution by

recent climate change to the event anomaly is very small and negative in EC-Earth ($-2\%$), CESM and MIROC agree on a small but positive contribution (7% and 10%, respectively, Fig. 7d). All models show a negative contribution of the ocean, which is most notable in CESM (around $-25\%$), while in EC-Earth it is smaller ($-7\%$) and almost negligible in MIROC ($-2\%$). This is in line with the La Niña conditions that prevailed from mid-2010 to early 2012 and then remained neutral for the rest of 2012 and during 2013 (NOAA Climate Prediction Center, 2022) as well as with the findings by Lewis and Karoly

(2013). For EC-Earth and CESM the contribution by the atmospheric circulation is larger than by soil moisture, whereas for MIROC it is the other way around. It was also found by King et al. (2014) that the dry conditions were an important driver of the heatwave. The ratio of atmospheric circulation contribution to soil moisture contribution is 60:40 for EC-Earth, 55:45 for CESM and 40:60 for MIROC. For MIROC, the individual ratios from the two disentangling approaches both agree that the contribution of soil moisture is larger than the contribution of the atmospheric circulation to the event anomaly whereas for the

other two models the individual ratios reflect that the contribution by the two main drivers of the Australian heatwave is equal to slightly circulation-dominated according to the experiments (Fig. A8d). This may reflect that the warm bias simulated by MIROC is alleviated significantly with soil-moisture constrained experiment (Fig. 4).

    Overall, the three models mostly agree on the relative contribution of atmospheric circulation vs. soil moisture to the TX anomaly during four recent heatwaves. For the heatwaves of 2010 in Russia and 2015 in Europe, all models show that the

atmospheric circulation plays the most important role. For the U.S. heatwave 2012, the models agree that soil moisture conditions are about as important as atmospheric circulation for driving the TX anomaly. For the Australian heatwave of 2012/2013, two models show that atmospheric circulation is more important whereas one model shows that the soil moisture contribution was largest. All models agree on a small role of climate change in driving the TX anomaly during the 2010 Russian heatwave and the 2012/2013 heatwave in Australia. However, for the 2015 European heatwave and the U.S. heatwave in 2012, the role

of climate change differs between the models, being largest for MIROC and smallest for EC-Earth. The role of the ocean is small for the heatwaves of 2010 in Russia, 2015 in Europe and 2012 in the U.S. For the 2012/2013 heatwave in Australia, all

models agree that the role of the ocean is negative – thus not enhancing the heatwave – however, the models disagree on the magnitude, with CESM being the only model displaying a notable contribution by the ocean.

## 5.2 Relative contribution of atmospheric circulation and soil moisture to episodes of anomalously warm temperatures

In the following, we analyse the role of atmospheric circulation and soil moisture in driving the occurrence of warm spells during 1982–2015/2016 (2015 for MIROC and 2016 for the other two models). The disentangling method is the same as used previously in Sect. 5.1. Warm spells are defined grid point-wise as time periods during the local summer season where daily mean temperature anomalies in ERA-Interim exceed 1.5 standard deviations of the 1982–2010 climatology for at least three consecutive days. A 7-day running mean is applied to the years 1982–2010 from ERA-Interim before computing the
daily climatology and standard deviation. The local summer season is defined as the three hottest consecutive months (from ERA-Interim) for each grid point. The threshold of 1.5 standard deviations was chosen such that most regions of the world actually experience events. However, using 1 or 2 standard deviations as threshold leads to very similar results (not shown). The identified warm spells are categorised into events of 3–5 days, 6–13 days and 14 days and longer. The choice of categories was made to separate events lasting a few days from week-long events and very long-lasting events of two weeks and more.
Also, the choice was made to obtain a reasonable sampling size for each category. The warm spells based on ERA-Interim are analysed by taking the same dates (calendar year and days of the year) in the experiments. First, the same dates are analysed for the fully constrained (AF_SF) experiment to determine the "model truth" for each event and model (using the ensemble mean for MIROC, which has five simulation runs). Then, the contribution of the drivers is disentangled according to Fig. 2. One or five simulation runs (over the years 1982–2015/2016) are used, depending on how many were available per model and
experiment. The mean temperature anomaly of each event category and experiment (averaged over all events and simulation runs) is used to disentangle the relative contribution of the atmospheric circulation and soil moisture conditions grid point-wise.

The agreement among models is very high for all spell lengths (Fig. 8). The grid points where soil moisture contributes one third or more to the warm spells, agree well with the regions of high soil moisture-temperature coupling (Koster et al., 2004; Miralles et al., 2012). With increasing spell length, the contribution of soil moisture becomes more important, for example in
the U.S. Midwest, Eurasia and northern Australia. Further, with longer spell length there is a growing proportion of total soil moisture contribution as can be seen by the increasing percentage of soil moisture dominance for all models in Fig. 8. This shows the growing relative importance of the land surface-atmosphere coupling for long-duration events.

The analysis also reveals that warm spells of 14 days and longer with a magnitude of more than 1.5 standard deviation do not occur often or in many regions of the world. The result for eastern Europe, for example, can be traced back to the Russian
heatwave in 2010. For tropical regions like Amazonia or very dry regions like the Sahara, it is not always possible to disentangle the relative contributions of atmosphere and soil moisture. This occurs because the computed differences can become negative if the less constrained experiments have a higher temperature anomaly than the more constrained experiments on average. The affected grid cells are masked out in white in Fig. 8.

It has to be noted that the analysis method takes into account the temporal persistence of warm spells but not temporal
correlation as for example the time-lagged effect of dry springs on hot summers (e.g. Hirschi et al., 2011; Quesada et al.,

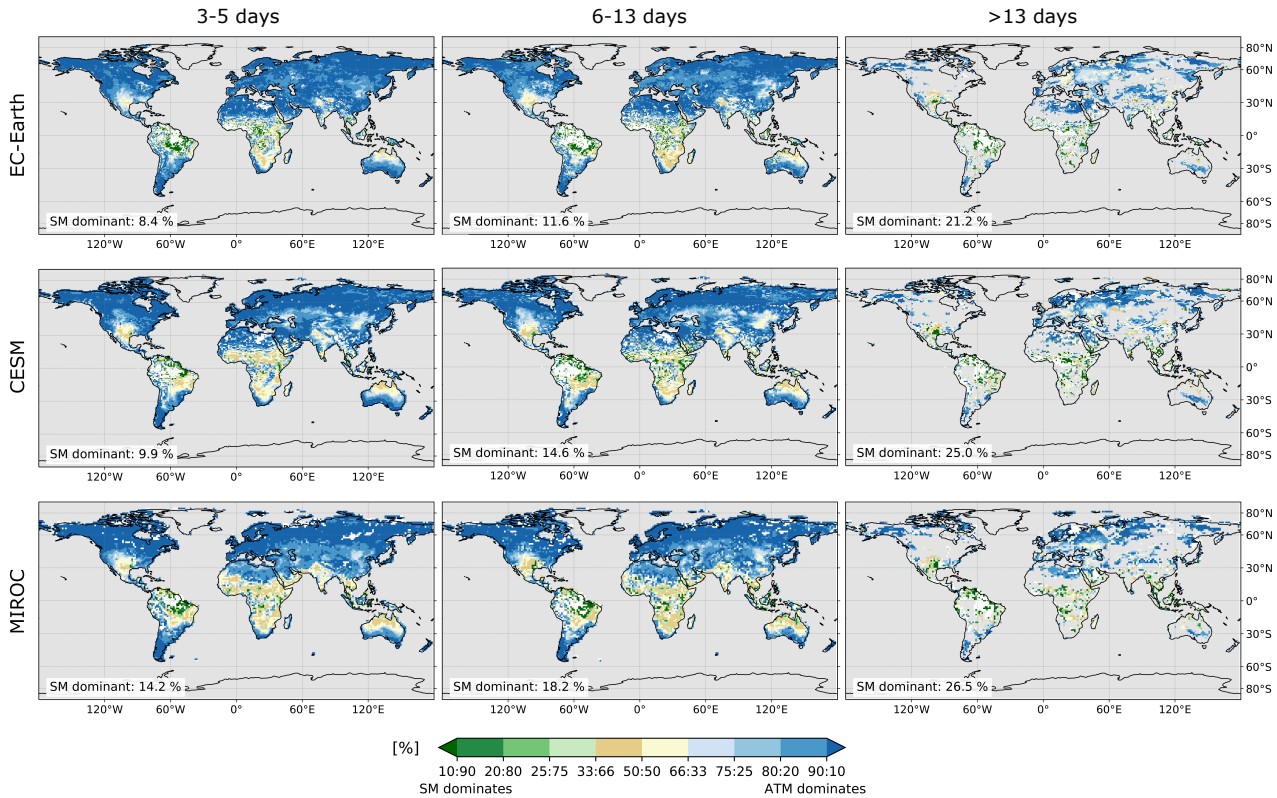

**Figure 8.** Contribution of atmospheric circulation (ATM) vs. soil moisture (SM) to warm spells during the local summer season where daily mean temperature anomalies exceed 1.5 standard deviation from the ERA-Interim 1981–2010 climatology. The local summer season is defined as the hottest consecutive three months (from ERA-Interim) for each grid point. The two approaches to compute SM vs. ATM contribution are merged giving equal weight to both. Events are categorised into spells lasting 3–5 days, 6–13 days and 14 days and longer. Ocean grid points, Antarctica, Greenland and Iceland are masked out in grey using the Greenland/Iceland ("GIC") region from the AR6 reference regions for the latter two. Grid points where no events where identified are also masked out in grey. Grid points where the contributions could not be determined (see text) are masked out in white. In the lower left corner the grid points where the SM contribution dominates over the ATM contribution (>50%) are given as area-weighted percentage with respect to all valid grid points.

2012). Furthermore, the events are only identified grid-point wise and not as spatially coherent patterns, as they would occur naturally. This is responsible for some of the noise in the patterns.

## 6 Conclusions and outlook

The ExtremeX experiment is a multi-model intercomparison project designed to study processes contributing to the occurrence and intensity of extreme events. ExtremeX currently consists of simulations with three ESMs: EC-Earth3, MIROC5 and CESM1.2. Five experiments were run with all models with one or more of the models' components being constrained. SSTs and sea ice coverage fractions are prescribed in all experiments. A grid-point nudging approach is used to constrain the modeled horizontal winds in the atmosphere and soil moisture prescription is used to constrain the land surface.

Although the constrained experiments capture the temporal evolution and magnitude of temperature anomalies well during recent heatwave events, climatological biases in temperature and precipitation remain in the experiments. This is the case both for experiments with either nudged atmospheric circulation patterns and/or prescribed soil moisture conditions. In some cases, biases are enhanced or even change sign in the constrained experiments (Fig. 3, Fig. 4 and Fig. 5). Comparing the location and magnitude of the climatological biases reveals that the patterns and sign of the biases often remain and the magnitude is only marginally reduced. This agrees with findings by Wehrli et al. (2018) for atmospheric nudging in CESM. The results suggest that the biases are caused by other processes such as cloud and precipitation formation, convection and interactions of the land surface and the ocean with the atmosphere, or also land surface parameters. It is also likely that none of these other elements is the sole explanation for the climatological biases, but rather their interaction, including atmospheric circulation and soil moisture (dynamics) interactions.

Despite the biases in mean climatology, the experiments with constrained atmosphere and soil moisture can accurately reproduce temperature anomalies during and prior to heatwaves (Fig. 6). This is found for all models and supports the results by Wehrli et al. (2019) for CESM. This result implies that alternatively bias correction could be used to improve the representation of extreme events in the models instead of analysing anomalies as is done here. The presented set of experiments can be used for extreme event analysis as long as the atmospheric circulation and/or soil moisture are major drivers of the event. The experiments are not ideal if the focus is on the role of the ocean, because the ocean is prescribed. For events that are mainly ocean-driven, we would recommend a setup with interactive ocean experiments to compute the ocean contribution more accurately. This would apply for example to extreme events (i.e. droughts, heatwaves, floods) that are strongly driven by the El Niño Southern Oscillation and other coupled ocean-atmosphere phenomena such as the Indian Ocean Dipole or the Pacific Decadal Oscillation. Nevertheless, here we derive the potential ocean contribution by comparing the anomaly to non-event years. The present study disentangles the role of atmospheric circulation vs. land surface processes for temperature anomalies. Therefore, additivity of the different contributions is assumed. This is inspired by the study by Kröner et al. (2017) for summer climate in Europe. Following the disentangling in Wehrli et al. (2019), experiments with constrained soil moisture (AI_SF) and with nudged atmosphere and soil moisture constrained to climatological values (AF_SC) are used along with the control (AI_SI) and fully constrained (AF_SF) experiments. The experiment with nudged atmosphere and interactive soil

moisture (AF_SI) leads to similar temperature anomalies during heatwaves like the AF_SF experiment. Atmospheric nudging

strongly constrains land surface conditions due to the control on available moisture and because in AF_SF ERA-Interim is used to derive the target soil moisture that is prescribed, similar land surface conditions result in both experiments. Hence, AF_SI is not considered in the disentangling procedure (see Fig. 2). To have more robust results, two disentangling approaches are considered like in Wehrli et al. (2019). Both approaches tend to produce similar results indicating that in a first order assumption the contributions can be treated as additive. Nevertheless, it has to be noted that disentangling causality in a coupled system

always comes with limitations. Differences between the approaches show nonlinearities in the responses due to feedbacks.

     TX anomalies during four recent heatwaves are attributed to their physical drivers and to climate change. The four events considered are: the 2010 Russian heatwave, the 2012 heatwave in the U.S., the Australian heatwave of 2012/2013 and the European heatwave in 2015. Overall the models show good agreement on the role of the drivers. Recent warming (since 1982–2008) is found to positively contribute to the event anomaly for all events and nearly all models (not for the Australian heatwave

2012/2013 and EC-Earth). The largest contribution by recent warming is found for the U.S. heatwave 2012 (15%–23%) and for the European heatwave 2015, however for the latter event the three models agree less on the relative role of climate change (12%–34%). In the presented setup the ocean was not found to have a substantial role in driving any of the events considered. This could be due to the limited interaction between the ocean and the atmosphere due to the prescription of SSTs and sea ice or because the ocean was indeed not a driver of the events considered. For the Australian heatwave 2012/2013, the ocean

is found to influence the temperature anomaly negatively in CESM (-23%). This is in accordance with the cool to neutral phase of the El Niño Southern Oscillation (NOAA Climate Prediction Center, 2022). For all four heatwaves the models show that both atmospheric circulation and land surface conditions significantly contribute to the event anomaly. For the Russian heatwave 2010 and the European heatwave 2015, atmospheric circulation is the dominant driver with land surface conditions playing a secondary but still important role. Yet, for the U.S. heatwave 2012, soil moisture is about as important as atmospheric

circulation and for the Australian heatwave 2012/2013, one model shows that soil moisture is the most important driving factor and the other two models show that the two physical drivers are about equally important. Note that, by design, the ExtremeX framework does not give information on which of the drivers is the initial source of the heatwaves since the constraining of the model components was carried out for the whole simulation period and events were analysed using contemporaneous anomalies from the experiments.

The ExtremeX experiments also allow a general assessment of the respective contributions of circulation anomalies vs. soil moisture conditions for warm spells. The results are very similar for the three ESMs showing that the models generally agree on the representation of extreme events and the driving processes behind these events. Warm spells of at least 3 days length are assessed grid point-wise and show that soil moisture is responsible for around one third to half of the temperature anomalies in transitional and tropical climate zones (Fig. 8). The regions identified resemble the regions of strong soil moisture-temperature

coupling highlighted by Miralles et al. (2012) for observational data and Seneviratne et al. (2006) for global climate models. Both studies additionally identify southern Europe and Eurasia as regions of strong soil moisture-temperature coupling which is, however, not confirmed from the results presented here. Nevertheless, in regions where spells of at least two weeks can

occur – like in Eurasia – soil moisture is more important for these longer events than for shorter events, driving up to one third of the temperature anomaly.

This study expands the mechanistic analysis of recent heatwaves by Wehrli et al. (2019) using three Earth System Models. The results for warm spells at the grid point level and for the four heatwaves suggest that both circulation patterns and soil moisture anomalies substantially contribute to the occurrence of heat extremes, which is consistent with Wehrli et al. (2019). Soil moisture effects are particularly important in the tropics, monsoon regions, and the US Great Plains, while circulation anomalies tend to dominate in other regions of the extratropics. These results can help to shed light on processes that need to

be better taken into account in weather predictions and climate projections. For instance, the important role of soil moisture conditions for extremes suggests that soil moisture monitoring and initialization could substantially improve forecasting of weather extremes in several regions.

*Data availability.* Simulation data from the models is available from https://data.iac.ethz.ch/Wehrli_et_al_2022_ExtremeX/

*Author contributions.* KW, MH and SIS designed the experiments with input from OM and RV. FL ran the EC-Earth3 simulations with technical help from FS, WM and PLS. HS ran the interactive and atmosphere nudged simulations with MIROC5. DT and HK ran the soil moisture nudged simulations with MIROC5. KW ran the CESM1.2 model simulations with technical support by MH. KW analysed the results from all models. KW, FL, MH, HS, DT, HK, DC, WM, OM, RV and SIS contributed to the discussion of results. KW prepared the manuscript with contributions from all co-authors.

*Competing interests.* The authors declare that they have no conflict of interest.

*Acknowledgements.* MIROC5 simulations were contributed by NIES Japan and University of Tokyo. EC-Earth3 simulations were contributed by VU Amsterdam and KNMI. CESM1.2 simulations were contributed by ETH Zurich. The authors thank the editor, Paul Dirmeyer and one anonymous reviewer for their valuable comments and suggestions to the manuscript. The authors also thank Dominik Schumacher for his feedback on the revised manuscript. KW and SIS acknowledge funding from the European Research Council (ERC) ("DROUGHT-HEAT" project, Grant 617518). FL, DC and FS acknowledge VIDI-award from Netherlands Organisation for Scientific Research (NWO) (Persistent Summer Extremes "PERSIST" project). HS was supported by the Integrated Research Program for Advancing Climate Models (JPMXD0717935457). HK acknowledges the National Research Foundation of Korea (NRF) grant funded by the Korea Government (MSIT)(2021H1D3A2A03097768 and NRF-2018R1A5A7025409). WM is supported through the Swedish strategic research area Mod-Elling the Regional and Global Earth system (MERGE). The MIROC5 simulations were performed by using Earth Simulator in JAMSTEC and the NEC SX in NIES. KW, MH and SIS acknowledge the support of Matthieu Leclair and Benoit P. Guillod in the design of the experiments and advice during the course of the project. This study uses the LandFlux–EVAL merged benchmark synthesis products of ETH Zurich produced under the aegis of the GEWEX and ILEAPS projects (http://www.iac.ethz.ch/url/research/LandFlux-EVAL/).

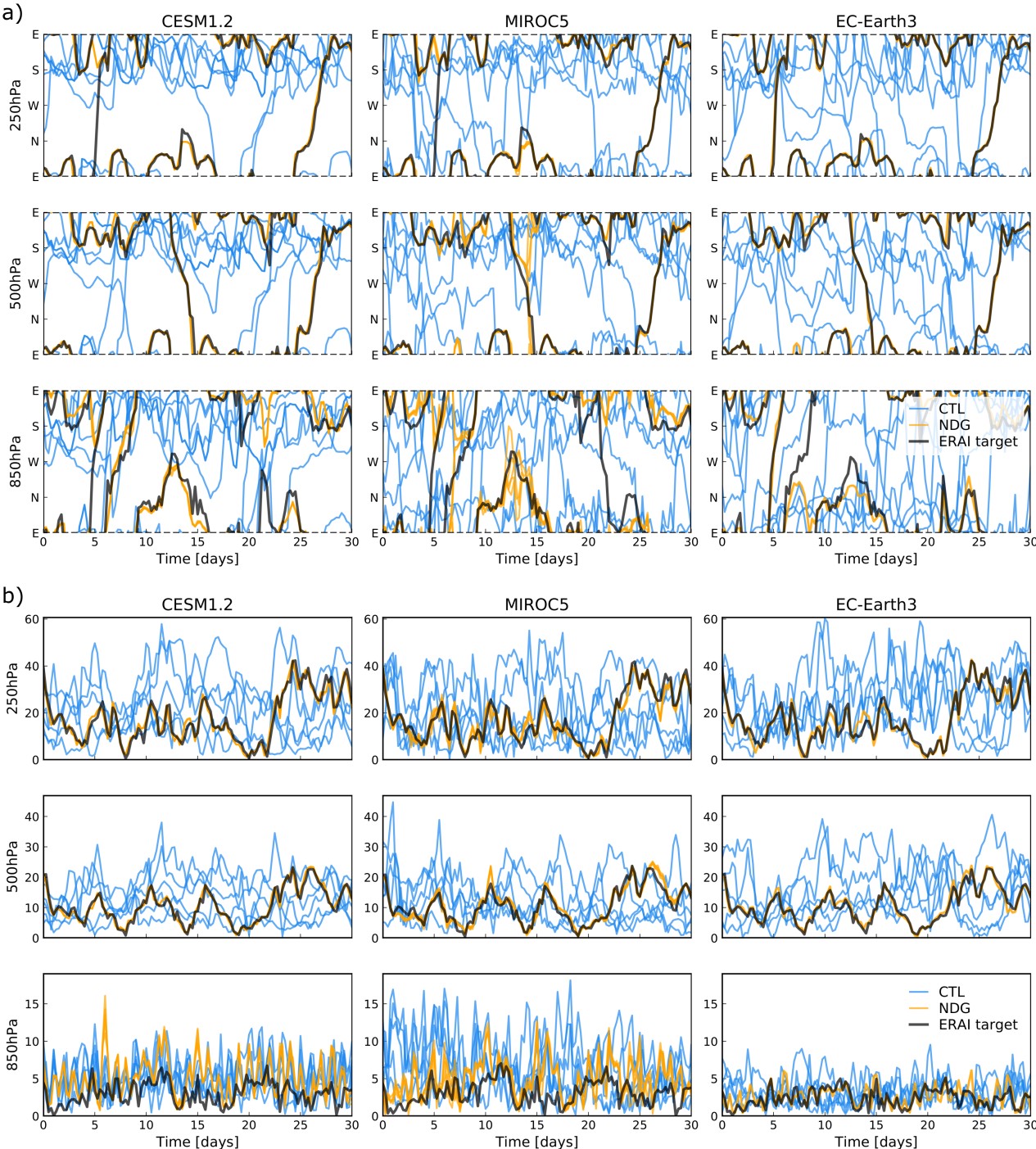

**Figure A1.** Comparison of winds for the experiments with nudged atmosphere (NDG, corresponds to AF_SI) and free atmosphere (CTL, corresponds to AI_SI). Shown are 6-hourly wind direction (a) and wind speed (b) during one month (June 2000) for one grid point over the Alps. The winds from the models were interpolated to 250 hPa, 500 hPa and 850 hPa. The winds from ERA-Interim (ERAI target) were interpolated to the same pressure levels and the model resolution for each model.

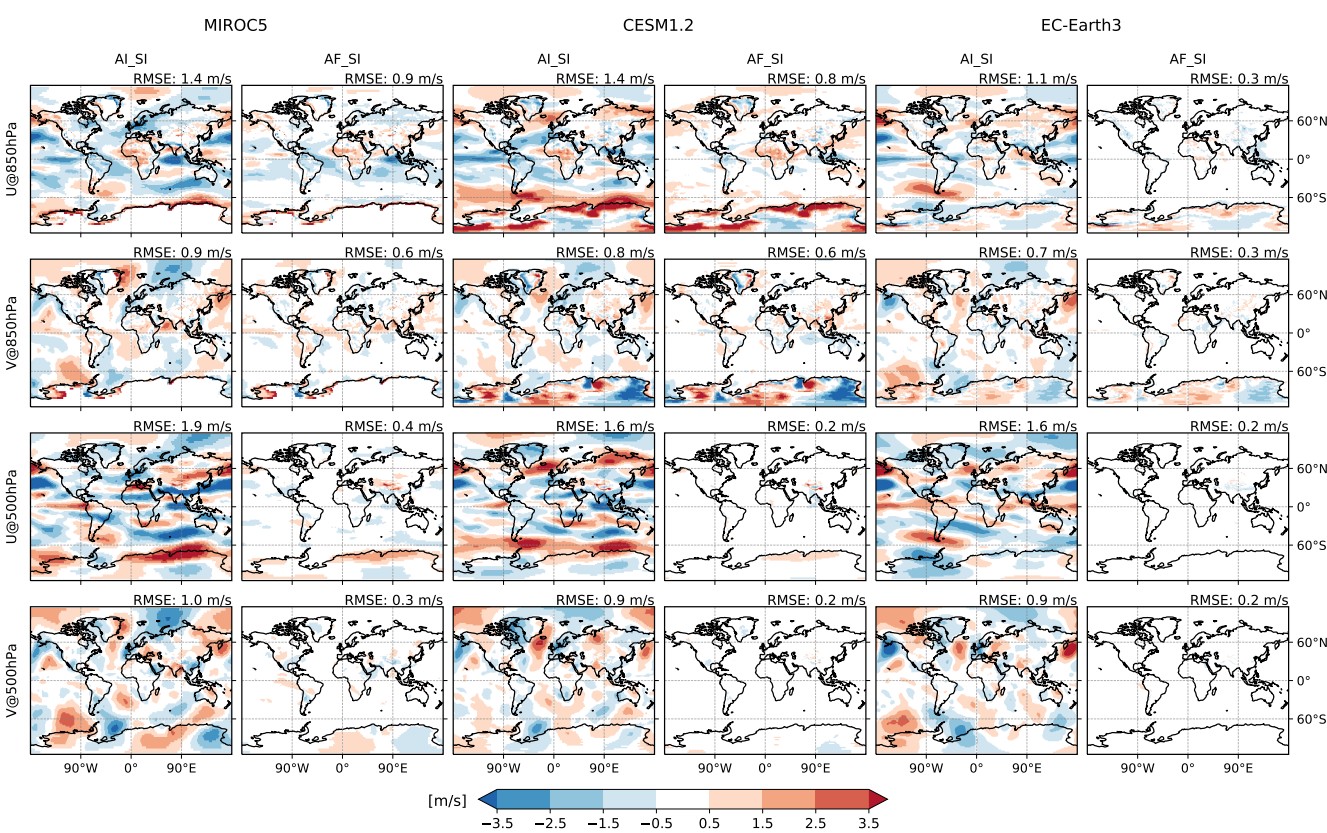

**Figure A2.** Bias of the zonal (U) and meridional (V) wind components at 850hPa and 500hPa for the experiments with nudged atmosphere (AF_SI) and free atmosphere (AI_SI), showing the ensemble mean where multiple simulations are available. Shown is the average over 6-hourly wind fields for one month (June 2000). The winds from the models were interpolated to 500 hPa and 850 hPa. The winds from ERA-Interim were interpolated to the same pressure levels and the model resolution for each model.

## AR6 reference regions and study regions

| Abbrev. | Name | Abbrev. | Name | Abbrev. | Name |
|---------|------|---------|------|---------|------|
| GIC | Greenland/Iceland | NEU | N.Europe | RFE | Russian-Far-East |
| NWN | N.W.North-America | WCE | West&Central-Europe | WCA | W.C.Asia |
| NEN | N.E.North-America | EEU | E.Europe | ECA | E.C.Asia |
| WNA | W.North-America | MED | Mediterranean | TIB | Tibetan-Plateau |
| CNA | C.North-America | SAH | Sahara | EAS | E.Asia |
| ENA | E.North-America | WAF | Western-Africa | ARP | Arabian-Peninsula |
| NCA | N.Central-America | CAF | Central-Africa | SAS | S.Asia |
| SCA | S.Central-America | NEAF | N.Eastern-Africa | SEA | S.E.Asia |
| CAR | Caribbean | SEAF | S.Eastern-Africa | NAU | N.Australia |
| NWS | N.W.South-America | WSAF | W.Southern-Africa | CAU | C.Australia |
| NSA | N.South-America | ESAF | E.Southern-Africa | EAU | E.Australia |
| NES | N.E.South-America | MDG | Madagascar | SAU | S.Australia |
| SAM | South-American-Monsoon | RAR | Russian-Arctic | NZ | New-Zealand |
| SWS | S.W.South-America | WSB | W.Siberia | EAN | E.Antarctica |
| SES | S.E.South-America | ESB | E.Siberia | WAN | W.Antarctica |
| SSA | S.South-America | | | | |

**Figure A3.** Reference regions of the IPCC AR6 as defined in Iturbide et al. (2020). The event regions used in Sect. 5.1 are indicated in red.

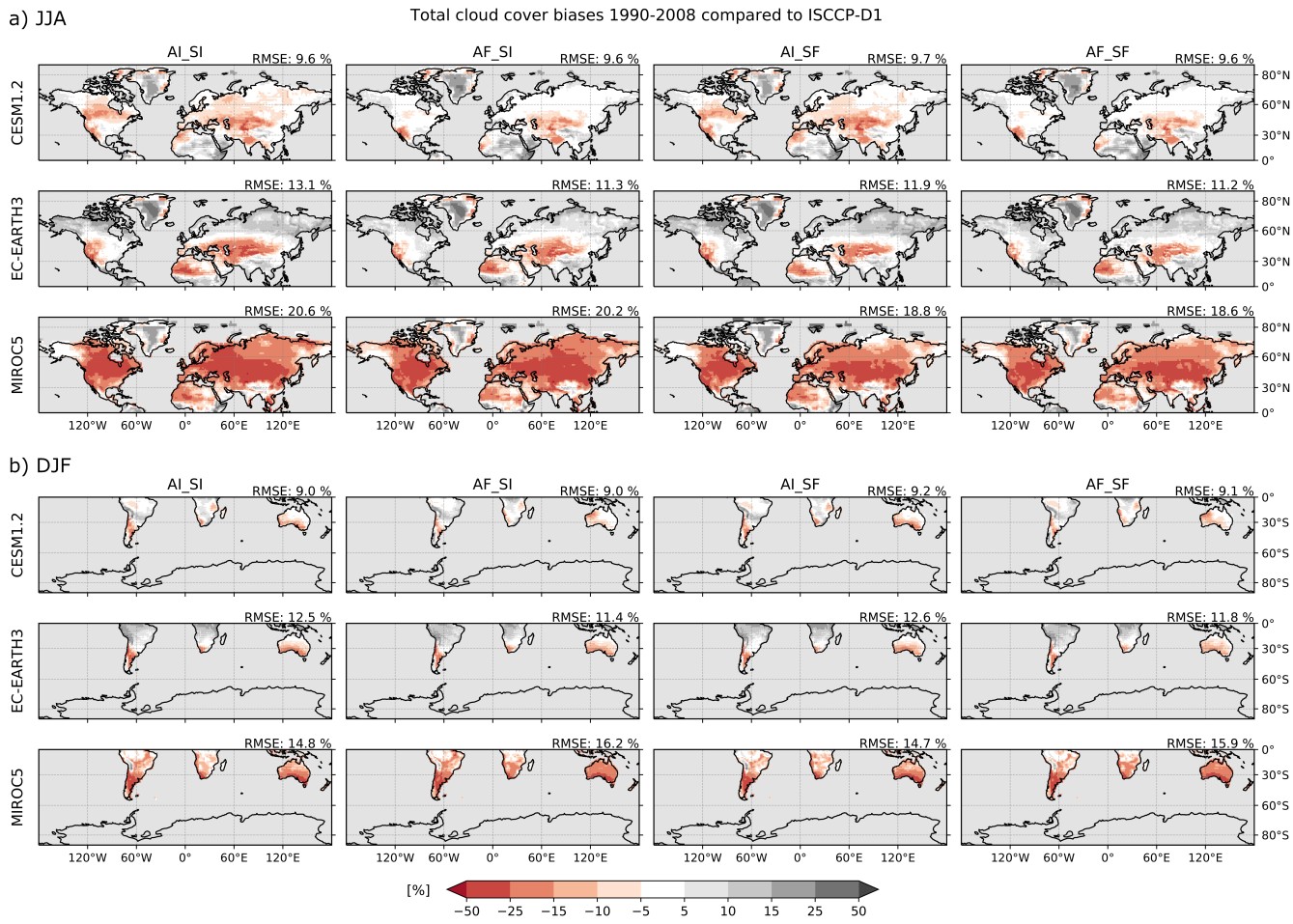

**Figure A4.** Bias in total cloud cover with respect to ISCCP-D1. Average over (a) JJA for the Northern Hemisphere and (b) DJF for the Southern Hemisphere. Ocean grid points and Antarctica are masked out. The RMSE averaged over all valid grid points of the respective hemisphere is given in mm per day in the upper right corner of each experiment and model.

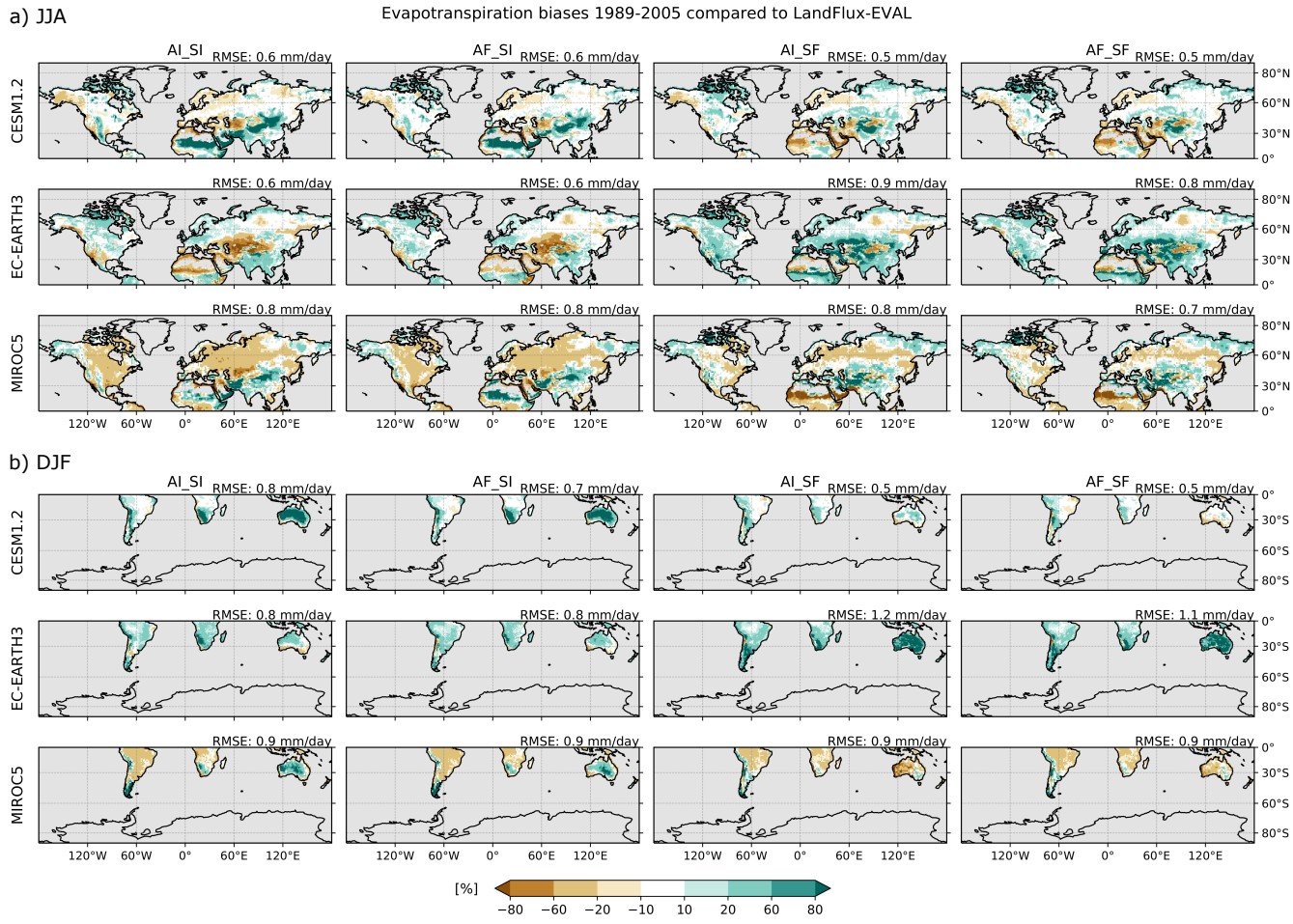

**Figure A5.** Bias in evapotranspiration with respect to LandFlux-Eval. Average over (a) JJA for the Northern Hemisphere and (b) DJF for the Southern Hemisphere. Masked out are grid points with a seasonal average of less than 0.1 mm evapotranspiration per day in the reference data set. Additionally, ocean grid points, grid points north of 75°N, Antarctica, Greenland and Iceland are masked out using the Greenland/ Iceland ("GIC") region from the AR6 reference regions for the latter two. The RMSE averaged over all valid grid points of the respective hemisphere is given in mm per day in the upper right corner of each experiment and model.

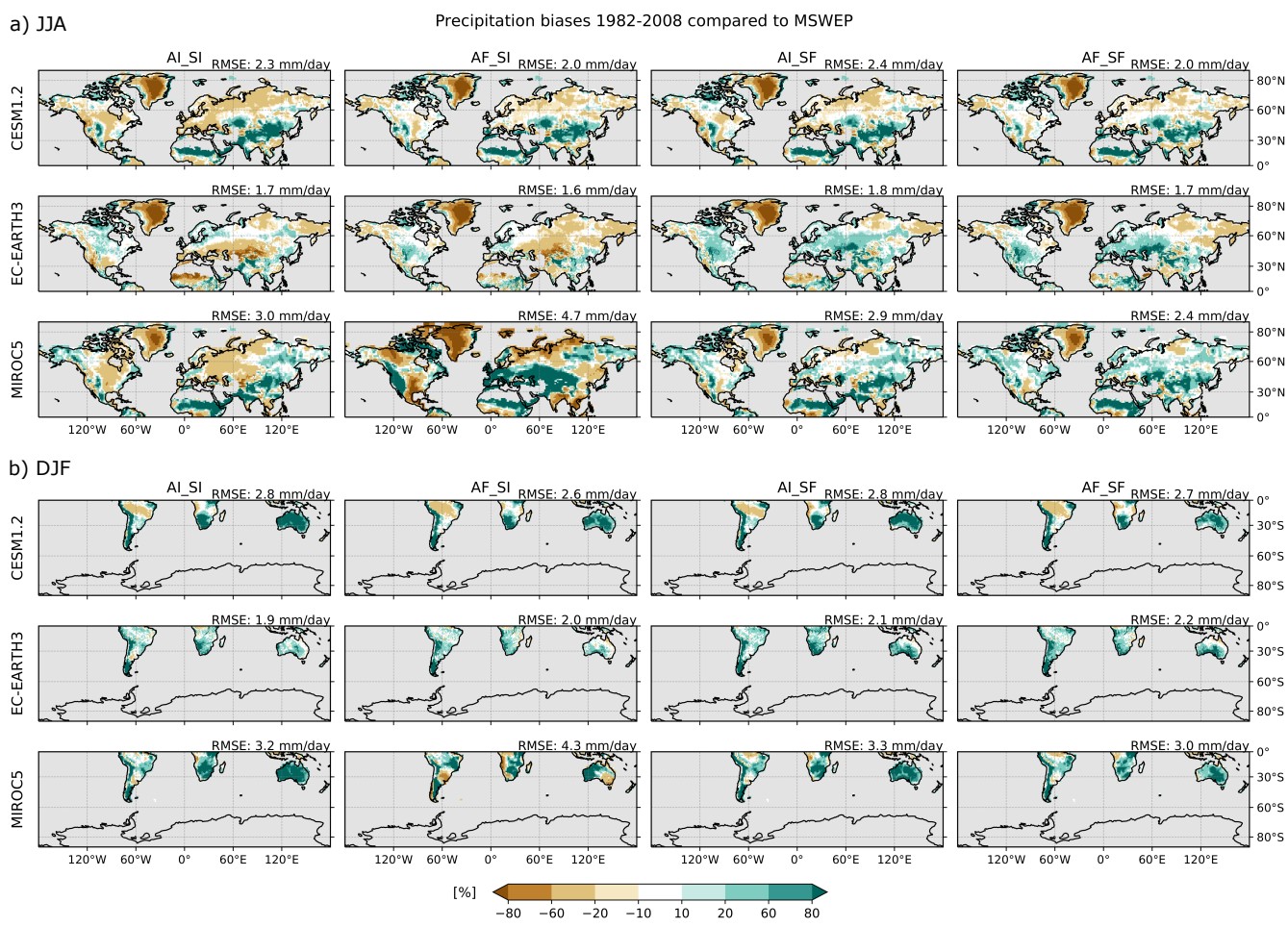

**Figure A6.** Same as Fig. 5 but using MSWEP as reference data set.

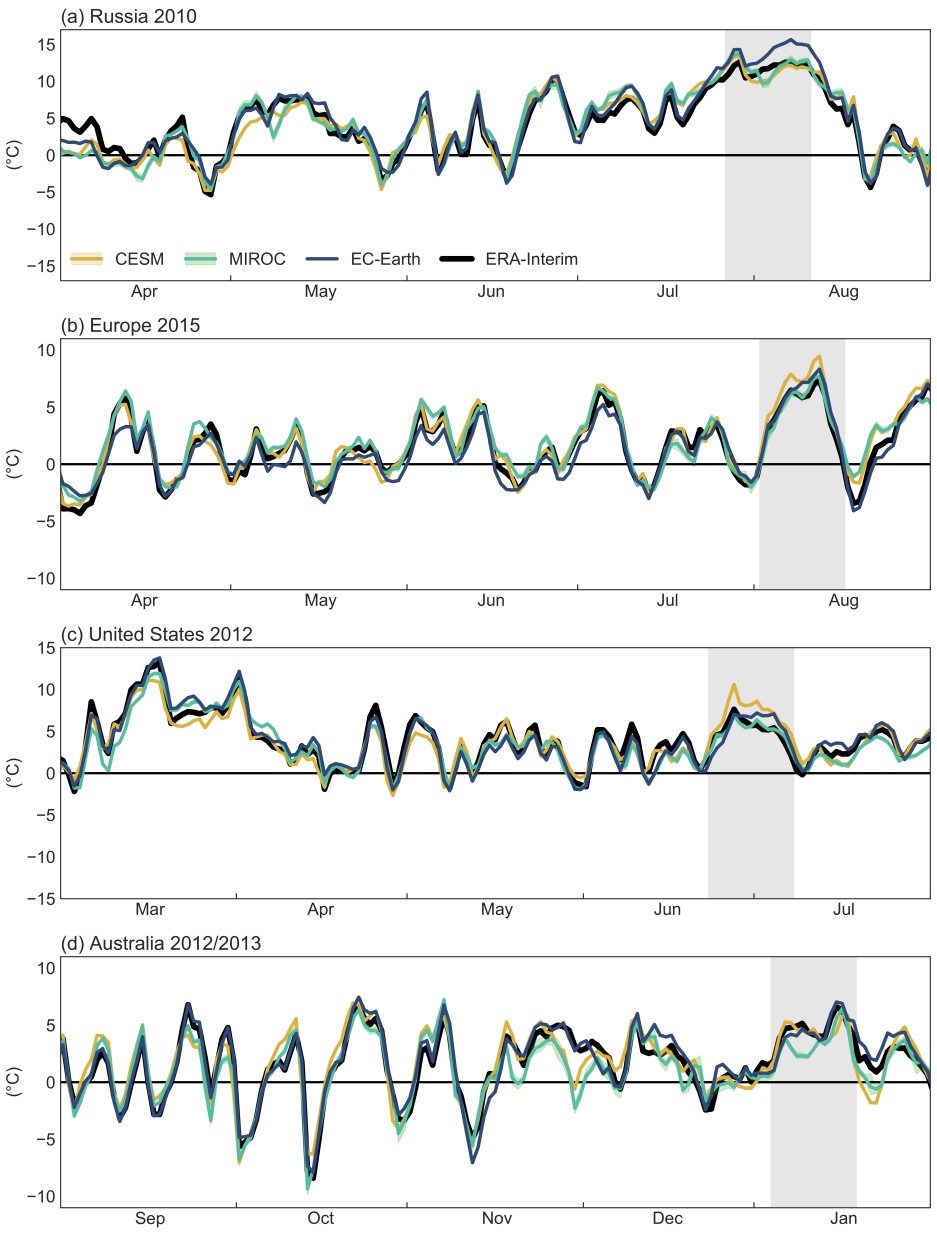

**Figure A7.** Same as Fig. 6 but for the nudging experiment (AF_SI). The shading shows the full ensemble spread and lines the ensemble mean (or single simulation for EC-Earth).

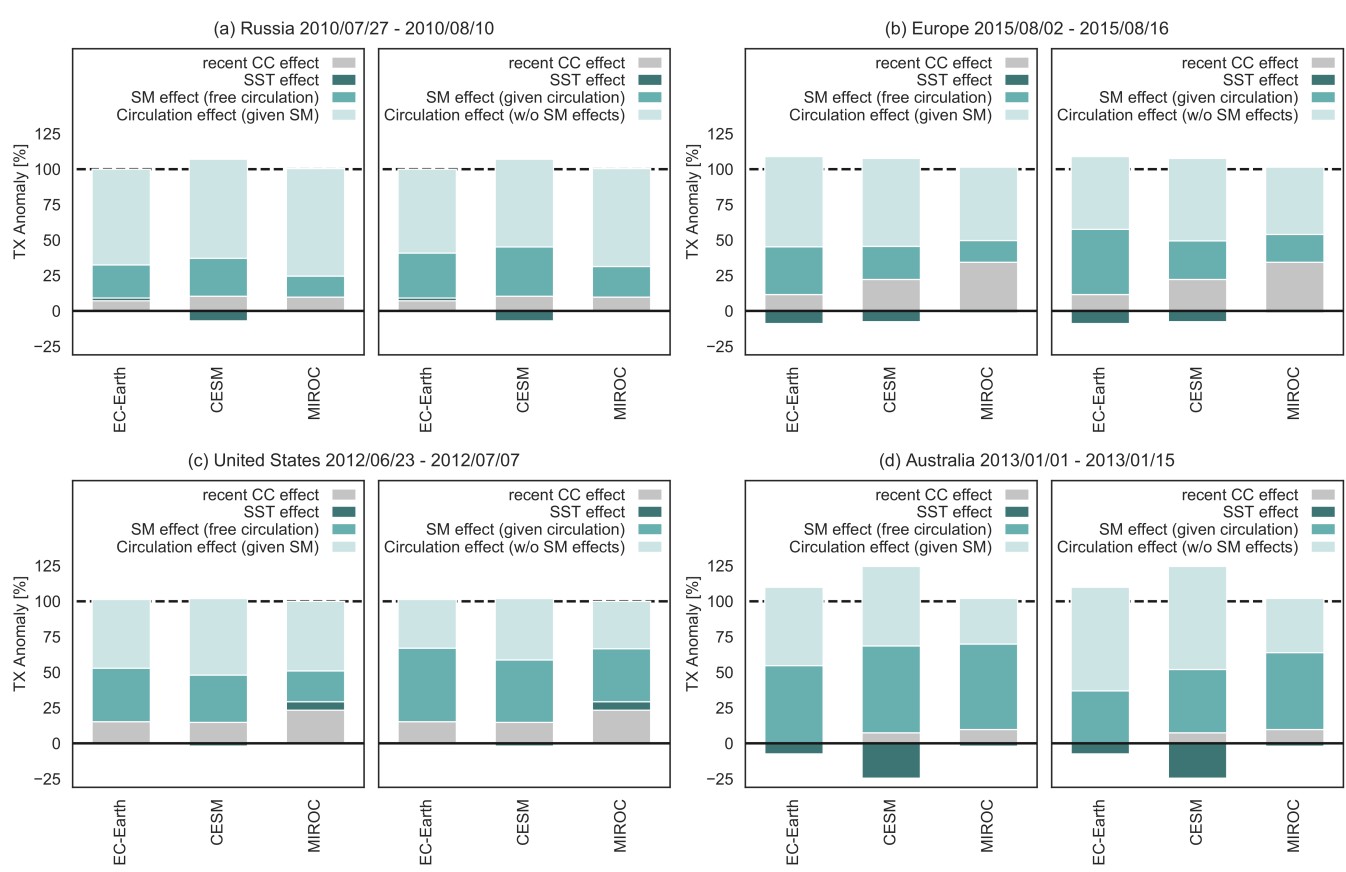

**Figure A8.** Same as Fig. 7 but showing the separate effects for the two approaches to compute SM vs. ATM contributions (left A, right B).

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
