# Peer review of "The ExtremeX global climate model experiment: Investigating thermodynamic and dynamic processes contributing to weather and climate extremes"

_Earth System Dynamics, 2021_

## Author Comment (AC1)

**The ExtremeX global climate model experiment: Investigating thermodynamic and dynamic processes contributing to weather and climate extremes**

By Kathrin Wehrli, Fei Luo, Mathias Hauser, Hideo Shiogama, Daisuke Tokuda, Hyungjun Kim, Dim Coumou, Wilhelm May, Philippe Le Sager, Frank Selten, Olivia Martius, Robert Vautard, and Sonia I. Seneviratne

**Reply to reviewer 1 (Paul Dirmeyer)**

The modeling experiments described here shed light on the various roles of land versus atmosphere in extremes, going a step or two beyond what was done in the 1990s and 2000s in the "Koster style" studies of those days. It is interesting, adds to our scientific knowledge of climate variability, and should be published after revision. I do not wish to remain anonymous - Paul Dirmeyer

We thank Paul Dirmeyer for his thoughtful and positive evaluation of the manuscript and the helpful suggestions. In the following we will give answers to the comments in blue.

General comments:

1. Realizing this may be difficult without redesigning and rerunning the simulations, but I long to see a bit more separation in the various drivers, e.g., in the atmospheric component, could the roles of dynamics (circulation) versus physics (radiation, clouds, precipitation) be separated? At the land surface, could drivers acting through the energy balance terms versus water balance be quantified separately? Others have delved more into the process level (e.g., https://doi.org/10.1029/2012GL053703), and having models in hand for sensitivity studies enables many possibilities. Likely for "future work", but I wanted to bring up this question.

2. I greatly appreciate the message of the paper regarding the role of compensating errors and tuning. There remains among many in the model development community a strange hope that "fixing" one component of an Earth system model (e.g., upgrading the LSM) will somehow solve other problems. But often it just serves to expose those problems even more as the balance of errors has been disturbed. This paper also shines a bit more light on this issue.

3. Mainly in §5.1 but also conclusions: The conventional wisdom is that persistent anomalies in the atmospheric general circulation (which may have various causes themselves) establish conditions for heat waves and/or droughts, and then land-atmosphere feedbacks can exacerbate or prolong them. Is there any way to diagnose (confront or confirm) this idea from these experiments? Can the role of climate change on this evolutionary sequence be investigated here? These analyses are co-temporal and do not seem to account for the evolution over time of heat wave events, although you do consider persistence. It seems the two "approaches" (A) and (B) get at this somewhat (e.g. L343-344) but it is somewhat elusive.

4. There are a couple of recent papers that are quite germane to ExtremeX, particularly the notion that heat waves have a mix of land and atmosphere (which may ultimately be traced to remote ocean) drivers: https://doi.org/10.1029/2020AV000283, https://doi.org/10.1002/asl.948.

Thank you for the thoughts and ideas for the manuscript. We also appreciate the reference for the two additional papers in the fourth comment. We will mention these relevant and very recent results in our introduction. Regarding a further separation of drivers this is certainly a very relevant question and we agree that it would be great to have future studies going in this direction. We agree with the reviewer that this would require a new experiment design and the additional simulations would be pout of scope for the present work. Regarding a separation of the processes at the land surface we could think of experiments similar to those in Teng et al. (2019), where heating anomalies (from a dry simulation) are imposed in the atmospheric model.

As mentioned in the third comment the experiments are co-temporal. Hence, the ExtremeX setup is likely not ideal to confront or confirm whether circulation anomalies establish conditions for heatwaves or droughts and then land-surface feedbacks kick in by prolonging the events. Studies like Teng et al. (2019) and Martius et al. (2021) have shown that soil moisture anomalies can excite atmospheric circulation anomalies impacting the weather in other regions of the globe. Having experiments where the constraining of the soil moisture (or atmosphere) is confined to a certain period of time (and region) helps to isolate the processes and reduces other interactions. However, the soil moisture anomalies applied in the two studies mentioned would have to be created first which would likely be due to circulation anomalies. Going more deeply into this question would likely require dedicated case studies. Further, we think that the influence of climate change on the development of heatwaves/ droughts may be better investigated in fully coupled model simulations, potentially with a large ensemble to capture a sufficient number of events.

Reference:

Teng, H. Y., G. Branstator, A. B. Tawfik, and P. Callaghan, 2019: Circumglobal response to prescribed soil moisture over North America. J. Climate, 32, 4525–4545, https://doi.org/10.1175/JCLI-D-18-0823.1.

Martius, O., Wehrli, K., & Rohrer, M. (2021). Local and Remote Atmospheric Responses to Soil Moisture Anomalies in Australia, *Journal of Climate*, *34*(22), 9115-9131. Retrieved Dec 6, 2021, from https://journals.ametsoc.org/view/journals/clim/34/22/JCLI-D-21-0130.1.xml

Specific comments:

L75: Technical point: an ensemble of one is not an ensemble. It is just a single run.

This will be corrected.

Fig 1: It would be more clear to replot with the X-axis in a time dimension, e.g., label it as the e-folding (relaxation) time scale.

We think that for the manuscript it is more illustrative to keep the plot with the nudging intensity on the x-axis. However, we will add the formula used to compute the relaxation term to clear things up.

L111: Change "allows to isolate" to "allows isolation of".

This will be corrected.

L124-125: Which models nudged and which replace soil moisture states? And for those that nudged, what was the relaxation time scale?

Thank you for the question. It turned out that it was a misunderstanding among the modeling groups that in MIROC a soil moisture nudging was used. In fact, all models replace, hence prescribe, the simulated soil moisture. We will correct this throughout the manuscript, which will also simplify the terminology used.

L131: I think there was more than one version (combination of inputs) for the LandFlux-Eval data set for ET - which was used?

The mean from the merged ET synthesis product was used (hence their diagnostic, reanalysis and land surface model-based data sets). We will add this information to the description of the reference data sets.

§2.4: This would benefit from a schematic. Could you reproduce or recreate a figure based on Fig 1 of Wehrli et al. 2019? It would be very helpful. And doesn't differences in the results from approaches (A) and (B) shed light on the nonlinearities in the responses (evidence of feedbacks)?

A simplified figure based on the one in the 2019 paper will be added. Indeed, differences in the results following the two approaches show nonlinearities in the responses. We will add a sentence to mention this. The results from the two approaches were found to be qualitatively similar therefore we will not explore the differences.

L285-288: To this list should be added "unrepresented processes" in models, particularly those unresolved due to grid scale: non-hydrostatic atmospheric processes in coarse resolution models, unresolved mesoscale circulations, sub-grid surface heterogeneity.

We thank the reviewer for this suggestion and we will amend the list with processes unrepresented in models.

L288-289: Atmospheric modelers in particular are fixated on 500 hPa geopotential height errors as a metric of circulation fidelity.

The predecessor papers Wehrli et al., 2018 and 2019 looked into the 500 hPa geopotential height for nudged CESM simulations. We will look into this also for the other models and describe the results for all three models in the manuscript.

Figs 5, 6 and associated text in §5.1: "Midwest" as a region name does not sit well in the global context, as it is a subregion of the U.S. In the other three cases, "Russia", "Europe" and "Australia" do not designate those entire areas, but a portion within each. Thus, "Midwest" should be replaced with "U.S."

The name of the region will be changed according to the reviewer's suggestion. For the events analysed in section 5.1 we will add that the U.S. heatwave was "also known as the Midwest heatwave" since the names of the events were chosen to match with existing literature.

For Russia (line 315) and the U.S. (line 336), how do these areas overlap or intersect the AR6 designated areas? Neither Fig A2 nor any of the other map plots in this manuscript show latitudes and longitudes, so it is difficult to compare by eye.

Figure A2 from the manuscript will be replaced by Figure R1 shown here. We will also update the figures to show longitude and latitudes where possible. Some figures already show rather small maps and we will make sure to not lose information or readability of the figures.

[Figure]

**AR6 reference regions and study regions**

| Abbrev. | Name | Abbrev. | Name | Abbrev. | Name |
|---------|------|---------|------|---------|------|
| GIC | Greenland/Iceland | NEU | N.Europe | RFE | Russian-Far-East |
| NWN | N.W.North-America | WCE | West&Central-Europe | WCA | W.C.Asia |
| NEN | N.E.North-America | EEU | E.Europe | ECA | E.C.Asia |
| WNA | W.North-America | MED | Mediterranean | TIB | Tibetan-Plateau |
| CNA | C.North-America | SAH | Sahara | EAS | E.Asia |
| ENA | E.North-America | WAF | Western-Africa | ARP | Arabian-Peninsula |
| NCA | N.Central-America | CAF | Central-Africa | SAS | S.Asia |
| SCA | S.Central-America | NEAF | N.Eastern-Africa | SEA | S.E.Asia |
| CAR | Caribbean | SEAF | S.Eastern-Africa | NAU | N.Australia |
| NWS | N.W.South-America | WSAF | W.Southern-Africa | CAU | C.Australia |
| NSA | N.South-America | ESAF | E.Southern-Africa | EAU | E.Australia |
| NES | N.E.South-America | MDG | Madagascar | SAU | S.Australia |
| SAM | South-American-Monsoon | RAR | Russian-Arctic | NZ | New-Zealand |
| SWS | S.W.South-America | WSB | W.Siberia | EAN | E.Antarctica |
| SES | S.E.South-America | ESB | E.Siberia | WAN | W.Antarctica |
| SSA | S.South-America | | | | |

*Figure R1: Reference regions of the IPCC AR6 as defined in Iturbide et al. (2020). Red outlines show the study regions considered in Section 5.1.*

L353: You discuss results from MIROC, but what about the other two models?

The individual rations for MIROC were mentioned to highlight that both approaches lead to the conclusion that SM dominates over the atmospheric circulation contribution. However, we understand that it is confusing why the individual ratios for the other models are not mentioned and we will revise this paragraph.

Fig 7: There seems to a growing proportion of contribution from soil moisture as the anomaly periods grow longer (which would be reasonable, as locally soil moisture represents a slower manifold, a redder spectrum than tropospheric variables). It appears this could be easily quantified. Showing the area-weighted average of the metric in the figure (e.g., the SM-dominant percentage, averaged over unmasked areas only) in each panel would show a growing value with warm spell duration in each model, showing the growing relative importance of the land surface states for long-duration events (which would get at the "conventional wisdom" point above, to some degree).

We thank the reviewer for this suggestion and we will update the figure and description accordingly.

L419-420: Is this true? The atmospheric nudging is very weak in the lower troposphere, and other studies have shown the effect of land surface anomalies on the atmosphere is largely constrained to the boundary layer (e.g., https://doi.org/10.1175/1525-7541(2001)002%3C0329:AEOTSO%3E2.0.CO;2 ) except over elevated terrain where heating anomalies from the land surface can get into the upper troposphere directly (https://doi.org/10.5194/gmd-14-4465-2021).

As the reviewer mentioned, the effect of land surface anomalies on the atmosphere in general is local and constrained to the boundary layer. We will rephrase the lines in the manuscript to make clear that the present study does not disagree with this statement. In the second study mentioned nudging of the horizontal wind was used to initialize the model before perturbing the land surface temperature. In the present study horizontal wind is nudged for every model time step during the whole simulation period. Indeed, we found that there is only negligible variability between ensemble members due to the setup of the atmospheric nudging. This is not only true for horizontal winds in the free atmosphere as shown for CESM in Wehrli et al. (2018) but also for land surface conditions. We illustrate this in Figure R2 by showing the daily maximum temperature anomaly compared to the 1982-2008 climatology for the Russian heatwave as in Figure 5a) but for the AF_SI experiment instead of AF_SF.

[Figure]

*Figure R2: Daily maximum temperature anomaly compared to the 1982–2008 climatology for the nudging (AF_SI) experiment for the three models and for ERA-Interim (black line). The 15-day event period is highlighted in light grey. The shading shows the full ensemble spread and lines the ensemble mean (or single simulation for EC-Earth). The thick black line shows the values from ERA-Interim.*

For CESM and MIROC five members for AF_SI were available and for EC-Earth only one. Figure R2 shows that the variability between the ensemble members is very small for both CESM and MIROC. For April to August (time period shown) daily standard deviation between ensemble members varies from 0.02°C to 0.19°C (0.07°C averaged over the whole time period) for CESM and 0.07°C to 0.47°C (0.22°C on average) for MIROC. The AF_SI experiment also captures the temporal evolution of TX anomaly similarly well as AF_SF. In Figure R3 the daily maximum temperature anomaly for all experiments is shown. For CESM (top) the AF_SI and AF_SF experiments barely differ while they do for MIROC (middle) and EC-Earth (bottom). This agrees with the findings from Figure 2 in the manuscript that AF_SI and AF_SF for CESM have a very similar climatology (and hence RMSE), which is not found for the other two models. This is due to the differences in how soil moisture was prescribed in the models and we will discuss this point in the manuscript.

[Figure]

*Figure R3: Daily maximum temperature anomaly compared to the 1982–2008 climatology for all experiments per model. Shown is the year 2010 and values are averaged for the region considered for the Russian heatwave. From top down: CESM, MIROC, EC-Earth. The shading shows the full ensemble spread and lines the ensemble mean (or single simulation). The black line shows the values from ERA-Interim.*

All map figures: Since soil moisture as a climate driver has no meaning over (under) permanent ice, glacial areas like Greenland should be masked from the maps.

Yes, we agree and figures will be redone to mask out glaciated areas.

Code and data availability: This is not consistent with COPDESS / FAIR data standards to which EGU journals adhere. Public data and/or code repositories should be used and indicated with permanent hyperlinks.

We will make the relevant fields for the figures shown available with the revised manuscript.

---

## Author Comment (AC2)

**The ExtremeX global climate model experiment: Investigating thermodynamic and dynamic processes contributing to weather and climate extremes**

By Kathrin Wehrli, Fei Luo, Mathias Hauser, Hideo Shiogama, Daisuke Tokuda, Hyungjun Kim, Dim Coumou, Wilhelm May, Philippe Le Sager, Frank Selten, Olivia Martius, Robert Vautard, and Sonia I. Seneviratne

**Reply to reviewer 2**

This paper presents the ExtremeX set of climate model experiments, where in three Earth System Models the moisture and atmospheric circulation are systematically constrained (nudged) towards observation-based values, either separately or jointly. Mean surface temperature and precipitation biases across these different experiments are evaluated, and it is found that these biases do not generally become smaller as the models are more constrained. The ExtremeX experiments are then applied to quantify the degree to which four recent strong heatwaves can be attributed to (i) sea-surface-temperature anomalies, (ii) atmospheric circulation anomalies, (iii) soil-moisture anomalies and (iv) recent climate change (from a 1979-2008 reference period to the time of the four events occurring within 2010-2015). The attribution method is then also applied to a wider set of warm spells during 2010-2015. It is found that most of the heatwaves and warm spells studied are predominantly due to circulation anomalies, with soil-moisture anomalies playing a secondary but important role, especially in subtropical and tropical regions. Contributions from sea-surface-temperature anomalies and recent climate change are typically much smaller than the other two.

The findings of this study are interesting, and it is nice to see a co-ordinated experiment across three models which lends robustness to the results, which will inform future model applications such as seasonal forecasting. I therefore recommend this study for publication in ESD subject to the comments provided below. While the presentation is generally clear, some additional investment in the introduction will make the paper more easily accessible to a wider audience. I also think that the model evaluation section would benefit from a concrete example (case study) in addition to the more general discussion provided so far. The role of the ocean in the ExtremeX setup also needs to be clarified.

We thank the reviewer for the detailed lecture and the helpful comments and suggestions to improve the manuscript. In the following we will address the points raised by the reviewer point-by-point. Answers to the comments made are given in blue below.

General comments

1) Introduction

Having read the whole paper, and then re-read the introduction, I can follow it much better, but I think some additional explanations (and clearer signposting of contents that is already provided) would make the introduction easier to follow, especially for other readers like me who are not necessarily familiar with the predecessor papers of this study. More specifically, I recommend paying attention to the following points:

- Some key references are provided, for example in the first two paragraphs and the lead author's own papers (line 48), but the main findings of these previous studies should be discussed in greater detail, as well as remaining knowledge gaps and which of these gaps this study aims to close.
- The focus and objectives of this study should be made clearer, especially which sort of extremes are to be studied. Line 41 rather vaguely mentions "extreme weather and climate events", whereas in the research questions it then transpires that the interest is in heatwaves/warm spells, although location, extent and duration remain unspecified. Part of my initial confusion seems to be due to the fact that there are two main purposes of this study, namely to (i) introduce the ExtremeX experiments (which I understand have a range of different possible applications) and to (ii) identify the drivers of heatwaves and warm spells, which is the specific application in this study. This distinction should be made clearer.
- Briefly motivate how to get from the conceptual distinction of dynamic and thermodynamic processes to setting up model experiments with constrained soil moisture/atmospheric circulation.

We thank the reviewer for sharing his experience from reading the introduction and the very helpful recommendations on how to improve the understanding and readability. We will follow the advice given. Specifically, we will explain the results of the Wehrli et al., 2018 and 2019 predecessor papers as briefly but also as completely as possible. Further, we will make clear from the beginning that the purpose of the study is to introduce the experiments and that we will apply them to study drivers of four recent heatwaves and to identify globally for which locations warm spells are generally dominated by processes at the land surface or by atmospheric circulation. We will also motivate the constrained experiments on the paragraph on line 36 saying:

"The processes driving a specific extreme event and their relative importance can be studied in observation-based studies using multiple linear regression (e.g. Arblaster et al., 2014; Wang et al., 2016) or forecast sensitivity experiments (e.g. Hope et al., 2016; Petch et al., 2020). In climate model simulations the role of the drivers can be studied by constraining the processes in the ocean, the atmosphere or at the land surface, which allows to study the drivers in isolation (e.g. Fischer et al., 2007; Hauser et al., 2016; Jaeger and Seneviratne, 2011). In this study, …"

References:

Arblaster, J. M., Lim, E.-P., Hendon, H. H., Trewin, B. C., Wheeler, M. C., Liu, G., & Braganza, K.(2014). Understanding Australia's hottest September on record. Bulletin of the American Meteorological Society, 95, 37–41.

Wang, G., Hope, P., Lim, E.-P., Hendon, H. H., & Arblaster, J. M. (2016). Three methods for the attribution of extreme weather and climate events (*018*): Bureau of Meteorology.

Hope, P., Wang, G., Lim, E.-P., Hendon, H. H., & Arblaster, J. M. (2016). What caused the record-breaking heat across Australia in October 2015? Bulletin of the American Meteorological Society, 97(12), S122–S126.

Petch, JC, Short, CJ, Best, MJ, *et al.* Sensitivity of the 2018 UK summer heatwave to local sea temperatures and soil moisture. *Atmos Sci Lett*. 2020; 21:e948. https://doi.org/10.1002/asl.948

Fischer, E. M., Seneviratne, S. I., Vidale, P. L., Lüthi, D., & Schär, C. (2007). Soil moisture-atmosphere interactions during the 2003 European Summer Heat Wave. Journal of Climate, 20(20), 5081–5099.

Hauser, M., Orth, R., & Seneviratne, S. I. (2016). Role of soil moisture versus recent climate change for the 2010 heat wave in western Russia. Geophysical Research Letters, 43, 2819–2826. https://doi.org/10.1002/2016GL068036

Jaeger, E. B., & Seneviratne, S. I. (2011). Impact of soil moisture-atmosphere coupling on European climate extremes and trends in a regional climate model. Climate Dynamics, 36(9), 1919–1939.

2) Validation of experiments

In section 4.2 (roughly Lines 258-283), a general discussion is provided of the issues that can arise in the constrained experiments based on tuned fully interactive models. I don't disagree with this discussion, but it is a little unsatisfactory as it stands, and I think an example (case study), possibly in a new subsection 4.3, could help to illustrate some of these issues more clearly. A case in point already highlighted by the authors are the large summer precipitation biases seen in the MIROC5 AFSI experiment (Fig. 4) without, I believe, correspondingly large biases in clouds or evapotranspiration (Figures A3, A4). I suggest analysing this further, for example by evaluating the moisture budget (including circulation and transport) of the different experiments in a suitable study region. A possible example is WCA, where, remarkably, the precipitation bias changes sign and the RMSE increases from 0.53 to 4.3 mm/day from AISI to AFSI.

Thank you for the comment. This is a really interesting suggestion. Unfortunately, we do not have all necessary variables from all models for the moisture budget evaluation. As we write in the manuscript it is known that MIROC5 shows large biases of the atmospheric circulation for example in the North Atlantic stormtrack (Zappa et al., 2013). These issues have been and are being targeted in model development. Since the tuning of the model parameterisations compensates for the deficiencies in the circulation we think that the analysis of the moisture budget will not suffice to explain the issues and biases seen. The aim of the presented study is to introduce the ExtremeX models and experiments, the constraining methodologies used and to provide an example for an application of the framework. Hence, we think an analysis going more deeply into a discussion of the origin of biases would be out of the scope of this paper. However, we feel like the question brought up by the reviewer would be a great starting point for a future study dedicated to understanding model biases and making recommendations for model improvement.

3) Role of the ocean

I am unclear about the role of the ocean in the ExtremeX setup and for the results of this study. This is illustrated in the conclusions: In Line 413, the authors say "Thus, the presented set of experiments can be used for extreme event analysis as long as the atmospheric circulation and/ or soil moisture are major drivers of the event." This means that the role of the ocean must be small – a working assumption, or limitation of the approach. However, in Line 431 it is asserted that "The ocean was not found to have a substantial role in driving any of the events considered" – this reads like a result of this study and may be seen to be incompatible with the earlier statement. Please explain this more clearly.

We thank the reviewer for pointing this out. The experiments can also be used if the ocean is an important driver of the event under consideration. Some of the analysis, like for example the separation in circulation vs. soil moisture driven in Figure 7 would not make a lot of sense in that case. The experiments are also not ideal, if the focus is on the role of the ocean because the ocean is prescribed. If experiments with interactive ocean were available, the ocean contribution could be computed more accurately, which we would recommend for mainly ocean-driven events. In that case the experiment setup would require an additional ensemble of 100 simulations like AI_SI. We will rewrite the sentences on L413 and L431 to explain this.

Minor comments

4) Abstract

The last sentence about where soil moisture effects are important raises the expectation of a similar sentence for the circulation effects.

We will rephrase the last sentence to say: "Soil moisture effects are particularly important in the tropics, the monsoon areas and the Great Plains of the United States, whereas atmospheric circulation effects are major drivers in other mid- and high-latitude regions."

5) Line 19

What does "consistent" here refer to? Extreme and mean model biases? Or maybe the range of CMIP5 models?

Consistent refers to consistent across models. We will rearrange the sentence to make this more clear:

"For climate models used in the fifth phase of the Coupled Model Intercomparison Project (CMIP5) consistent biases can be found across models in the mean climatology of the lower atmosphere and land surface, for example temperature and precipitation …"

6) Line 48

"… by validating the forcing of the atmosphere and the land for the near-surface climatology." I did not understand this (before reading the paper). Please rephrase.

We will change this sentence to: "The presented work expands on previous work in Wehrli et al. (2018, 2019) by quantifying biases of the near-surface climatology for different constraining experiments and three models."

7) Line 52

Specify "overall model biases".

We will rephrase this to say "climatological model biases".

8) Table 1

Provide the number of ensemble members as three comma-separated values using a specified order of models, e.g., "5,5,1" for 5 members in CESM1.2, EC-EARTH3, and 1 member in MIROC5.

Thank you for the suggestion, we will do this in the revised manuscript.

9) Line 94

I suggest listing/explaining the different terminologies once upfront (forcing, constraining, nudging, relaxing) and then to stick to one choice for the remainder of the paper. "Constraining" seems to work well.

It turned out that due to a misunderstanding between the modeling groups the assumption was that different methodologies were used to constrain soil moisture. In fact, soil moisture was prescribed in all models. We will therefore use the terms "soil moisture prescription" and "atmospheric circulation nudging" throughout the manuscript and explain the terminology at the beginning of Section 2.2.

10) Line 139

Regarding the "additivity" – can this, or has this, been tested? Clarify briefly.

The additivity assumption has not been tested for the disentangling method presented here. However, it was based on the study by Kröner et al. (2017) where it was shown that it can be assumed that the contribution of the thermodynamic effect due to global warming, the lapse-rate effect and the large-scale circulation (as well as remaining effects) to the summer climate in Europe are additive. The assumption was tested for other seasons but not for other regions in that study. We think that disentangling method A and B giving similar results for all models is a further indication showing that in a first order assumption the contributions can be treated as additive. We will mention this in the manuscript.

Reference:

Kröner, N., Kotlarski, S., Fischer, E. *et al*. Separating climate change signals into thermodynamic, lapse-rate and circulation effects: theory and application to the European summer climate. *Clim Dyn* **48,** 3425–3440 (2017). https://doi.org/10.1007/s00382-016-3276-3

11) Line 141

Replace "analyses investigate" by "disentangling method determines".

We will change the sentence as suggested.

12) Line 172

Replace "The target data set" by "The prescribed target soil moisture" (if true).

We will change the sentence as suggested.

13) Line 192

Explain (or omit) "non-operational".

We will omit the expression. What is meant is that the model is not used for making actual (operational) weather forecasts or seasonal predictions.

14) Line 206

Replace "toward observations" by "toward reanalysis" (if true). Make this distinction throughout.

"Toward reanalysis" is correct, we will change that in the manuscript.

15) Figure 2

This figure nicely summarises the performance for different experiments and regions!

Thank you!

16) Figure 4

Explain the grey areas in the caption.

Ocean grid points and Antarctica are masked out in Figures 3 and 4. We will add this to the captions.

17) Line 292

"… nudging the atmospheric large-scale circulation and constraining the soil moisture results …" – The *and* seems key here as there can be substantial biases in the experiments where circulation and soil moisture are constrained individually. Please discuss if this is expected to impact the disentangling method.

There are substantial (climatological) biases in all experiments as we do not perform any bias-correction. Constraining the circulation or soil moisture individually can lead to larger biases than in the unconstrained setup as for example for precipitation in MIROC for the AF_SI experiment. In section 5 we are interested in temperature anomalies during specific events or warm spells in general. These anomalies are always computed with respect to the climatology of each experiment and model individually. Hence, climatological biases do not come into play here and do not impact the disentangling method. The section focuses on the magnitude of the anomaly and the temporal evolution during specific events. In fact, the nudging only experiment (AF_SI) already compares very well to temperature anomalies from ERA-Interim and even the constrained soil moisture experiment tends to show positive anomalies during the events examined. To make this clearer we will add figures like Figure R3 in the response to Paul Dirmeyer to the appendix.

18) Line 319

Previous work has suggested an important role of anomalous sea surface temperatures for the 2010 Russia heatwave (Trenberth and Fasullo 2012). This study finds that "CESM is the only model which shows a negative ocean contribution of around −7%, whereas the role of the ocean is negligible in the other models".

Does this mean that this study contradicts Trenberth and Fasullo 2012? Is there further evidence for or against in the literature?

Such context with the existing literature should be briefly discussed – also in the conclusions and for the other three events (see also comment 1).

There is other literature supporting a weak role of the ocean to the Russian heatwave like Dole et al. (2011) and Hauser et al. (2016). On the other hand, the Trenberth and Fasullo (2012) study is supported for example by the study of Martius et al. (2013) who link SST anomalies to atmospheric circulation conditions over the Asian continent leading to the Pakistan floods and the Russian heatwave. In that sense our results do contradict these studies. However, this has certainly also to do with differences in experimental setup as the findings here are based on simulations with a prescribed ocean. We will include more discussion of existing literature and comparison to our findings in the revised manuscript.

References:

Dole, R., Hoerling, M., Perlwitz, J., Eischeid, J., Pegion, P., Zhang, T., Quan, X.-W., Xu, T., & Murray, D. (2011). Was there a basis for anticipating the 2010 Russian heat wave? Geophysical Research Letters, 38, L06702. https://doi.org/10.1029/2010GL046582

Hauser, M., Orth, R., & Seneviratne, S. I. (2016). Role of soil moisture versus recent climate change for the 2010 heat wave in western Russia. Geophysical Research Letters, 43, 2819–2826. https://doi.org/10.1002/2016GL068036

Martius, O., Sodemann, H., Joos, H., Pfahl, S., Winschall, A., Croci-Maspoli, M., Graf, M., Madonna, E., Mueller, B., Schemm, S., Sedláček, J., Sprenger, M. and Wernli, H. (2013), The role of upper-level dynamics and surface processes for the Pakistan flood of July 2010. Q.J.R. Meteorol. Soc., 139: 1780-1797. https://doi.org/10.1002/qj.2082

19) Line 376

"The spells are analysed by taking the same dates in the experiments." – I don't understand this.

Warm spells are identified and categorised based on the ERA-Interim reanalysis. Then the same date (calendar year and days of the year) is analysed in the experiments. Hence, if at a certain location a warm spell (during the local warm season) lasts from August 12 to August 17 of a given year, temperature anomalies in the experiments during August 12 to August 17 of the same year will be used. We will clarify this in the manuscript.

20) Figure 7

What limits this application to events that last longer than ~2 weeks? Is this simply a question of sampling/ensemble sizes, or an inherent limitation of the disentangling method? Please discuss this briefly.

The choice of the categories used for warm spell lengths was motivated both by sampling size but also other considerations. The lower bound of three days was chosen due to the common definition of heatwaves lasting at least three days or longer. The separation between 5 and 6 days for categories 1 and 2 was made subjectively to separate events lasting a couple of days from events lasting roughly a week but it was also made to obtain a similar sample size. The last category for events lasting two weeks and more was introduced to have an additional separation for very long-lasting events like for example the Russian heatwave. Due to the small sample size (and some regions not showing events of this length) introducing more categories for even longer events would not make sense for the global analysis carried out here.

21) Figure 7

Say in the caption how the local warm season is defined.

The warm season is defined as the hottest consecutive three months (from ERA-Interim) for each grid point. We will explain this in the figure caption.

Reference

Trenberth, K. E., & Fasullo, J. T. (2012). Climate extremes and climate change: The Russian heat wave and other climate extremes of 2010. Journal of Geophysical Research: Atmospheres, 117(D17), n/a-n/a. https://doi.org/10.1029/2012JD018020

---

## Editor Decision (ED1)

Editors review Wehrli et al. ESD-2021-58

Dear author, co-authors,

First of all, sorry for the delay in this editors decision, due a misunderstanding, on your ms submitted for publication in ESD. Having read in detail your response to the reviews as well as the suggested revisiond of the manuscript, I thought that I could basically accept the revision for publication in ESD but then reading once more carefully the revised ms (including the one with track changes, note the differences in line numbers), I came still across quite some (revised) statements that were not optimally formulated/require further editing/triggered some additional questions. In addition, there were also some other issues being raised in reading the ms in terms of the overall context. Below you can find all these and would like you to handle these comments for my final decision on your paper.

Line 37: "and new, yet unseen,: not yet seen extreme intensities are appearing"

Something that has not been seen yet has also not appeared..... I would suggest here to use wording; "are anticipated"

Line 51: "predecessor paper", suggest to refer here to "previous study"

Line 58/59: "atmospheric nudging and/or prescribed soil moisture"

Line 60-61: " This nudging approach has previously been verified for CESM (Wehrli et al., 2018) also analyzing biases...."

Line 61: "Here, the same experiments are carried out for three ESMs". Now that it becomes much more clear how the presented study compare to the previous studies only having used the CESM system, it would be good to stress this: "Here, building further upon the studies with CESM, we here present the results of the same experiments but carried out for ...."

Line 64/65: "Other applications have not been tested so far and will be left to explore in future studies". Now that you have included this to address the reviewers comments, e.g., on more detailed analysis of the water balance but now this statement, especially mentioning "other applications"" is way to vague. Can you indeed indicate how the ExtremeX experiment results can be diagnosed in a different manner to further unravel the role of land surface processes versus circulation (and the oceans)?

Line 82: for consistency use the term "observation-based data"

Line 85: "All experiments prescribe SSTs, sea ice and vegetation.." would be good indicate here what sea-ice and vegetation characteristics are prescribed, cover? Or other properties?

Line 92-93; had to read this sentence multiple times: suggest to change to "...experiments both components are constrained prescribing soil moisture varying over time or prescribing soil moisture using a climatological soil moisture, respectively"

Line 95: "As ExtremeX was initiated in 2017, no longer time horizon was proposed". Should have commented on this earlier but what is meant with this? I don't get what you exactly want to express with this statement.

Line 103: "observation-based products" or "observation-based data"? be consistent

Line 113: "..to generally refer to the applied method of nudging the atmospheric large-scale circulation and prescribing soil moisture"

Line 130: "2.2.2 Prescribing Soil moisture"

Line 146: having referred before to the specific method of prescribing soil moisture and then introducing here in a very general way "2.3 Reference data sets", I would put soil moisture here under the same heading. But here you are introducing more the "Atmospheric circulation and hydrology data"

Then reading the introduction again of the three models that have been included in these ExtremeX experiments, I realized that I miss(ed) a motivation on the actual of these three models. What has been the main reason that these experiments include these three different modelling systems, is there a specific reason why these models have been selected and what would be the expected differences based on the main (different) features of these models? It would be good to shortly include this in the introduction of the paper.

Line 233: "...some variability in wind speed between the five members.."

Line 234: "..representation of the reference wind speed", to stress that this evaluation/validation step is mainly relying on comparison of wind speeds and not other meteo. properties

Lines 270/271: "This contradicts the initial intuitive assumption and suggests that no sole component of the model is responsible for the biases, and hence the latter cannot be corrected in isolation". I was somewhat puzzled by what you want to express by those last three words; I guess would like to express that the biased cannot be corrected by overcoming the causes of those simulated biases by improving the model representation of only one (in isolation) of the model components". If this is indeed what you mean, possibly rephrase (more explicitly)

Line 292: "Nonetheless, for EC-Earth AF_SF shows larger biases than AI_SI as the temperature biases introduced in AF_SF outweigh the biases corrected, and for precipitation, the biases remain of similar magnitude". What do you mean here with biases corrected?? Corrected for what?

Line 359: "which both found a weak role of the ocean in ....??" explaining the Russian heatwave or heatwaves in general?

Line 374: "by Duchez et al. (2016) finding that the SST patterns set important preconditions for the 2015 summer", same comment here; just for the 2015 summer in general or in explaining the 2015 European summer heatwave?

Line 399: "This is in line with the La Niña conditions that established in 2010 and 2011 and transitioned to cool to neutral during...", this statement does not read well and suggest to change to "This is in line with the La Niña conditions that prevailed in 2010 and 2011.." but want do you want to express with "transitioned to cool to neutral"?? a change to a more cool or neutrale state/contributiob bu La Nina to conditions in Australia?

Line 407: "whereas for the other two models the individual ratios are balanced to slightly circulation-dominated", another puzzling statement, I guess you want to express that "whereas for the other two models the individual ratios reflect that there might an even-to a slightly circulation-dominated contribution by the two main drivers of the Australian heatwave according to the experiments"

Line 418: "– thus not favoring a heatwave –" a would rather say "– thus not strongly enhancing the  heatwave– "

Line 442: "This shows the growing relative importance of the **land surface-atmosphere coupling** for long-duration events" (instead of only the state of the lanf surface, correct?)

Line 457: "Five experiments with varying levels of constraining were run with all models"; this is bad revised sentence, alternative; "Five experiments, with different sources of information to constrain these experiments, were run with all models"

Line 463: "even larger biases appear in the constrained experiments", alternative also given that there were already biases in the default experiments, "the biases are even further enhanced in some of the constrained experiments"

Line 468: "including atmospheric circulation and soil moisture (dynamics) interactions"

Line 474: "For events that are mainly ocean-driven, we would recommend a setup with interactive ocean experiments to compute the ocean contribution more accurately." Could you possibly shortly indicate/ hypothesize on extreme events that might strongly affected by ocean-atmosphere-driven process interactions. Would there be a role in heatwaves (in suppressing, or, enhancing) or other extreme events?

Line 476: "..the potential ocean contribution.."

Line 493: "The largest contribution **by recent warming** is found for the U.S. heatwave.."

Laurens Ganzeveld

---

## Author Response (AR2)

*ESD-2021-58: Response to editor*

**"The ExtremeX global climate model experiment: Investigating thermodynamic and dynamic processes contributing to weather and climate extremes"**

By Kathrin Wehrli, Fei Luo, Mathias Hauser, Hideo Shiogama, Daisuke Tokuda, Hyungjun Kim, Dim Coumou, Wilhelm May, Philippe Le Sager, Frank Selten, Olivia Martius, Robert Vautard, and Sonia I. Seneviratne

Dear author, co-authors,

First of all, sorry for the delay in this editor's decision, due a misunderstanding, on your ms submitted for publication in ESD. Having read in detail your response to the reviews as well as the suggested revisions of the manuscript, I thought that I could basically accept the revision for publication in ESD but then reading once more carefully the revised ms (including the one with track changes, note the differences in line numbers), I came still across quite some (revised) statements that were not optimally formulated/require further editing/triggered some additional questions. In addition, there were also some other issues being raised in reading the ms in terms of the overall context. Below you can find all these and would like you to handle these comments for my final decision on your paper.

Dear editor,

Thank you for carefully reading our manuscript and we appreciate your thoughts on the content. We have addressed the questions and comments raised and believe that this has augmented the quality of the paper further. The only additional change made to the manuscript is that paragraphs from the introduction have been rearranged such that we first state the research questions and scope of the paper, then mention possible further applications and finish by an overview of the content of the paper which now also indicates the section numbers. These changes in our opinion read better with the other changes made in the introduction. In the following, answers are given below the statements. Please also find with the re-submission the document with track changes (changes with respect to the previous revised manuscript from 28.02.22) and the newly revised manuscript itself.

Best regards
Kathrin Wehrli (on behalf of all authors)

Line 37: "and new, yet unseen, not yet seen extreme intensities are appearing"
Something that has not been seen yet has also not appeared..... I would suggest here to use wording; "are anticipated"

We thank the editor for this suggestion and the line was changed accordingly.

Line 51: "predecessor paper", suggest to refer here to "previous study"

done

Line 58/59: "atmospheric nudging and/or prescribed soil moisture"

done

Line 60-61: " This nudging approach has previously been verified for CESM (Wehrli et al., 2018) also analyzing biases...."

The sentence was changed as suggested.

Line 61: "Here, the same experiments are carried out for three ESMs". Now that it becomes much more clear how the presented study compare to the previous studies only having used the CESM system, it would be good to stress this: "Here, building further upon the studies with CESM, we here present the results of the same experiments but carried out for ...."

Thank you for this suggestion, we changed the line as suggested (except for cutting the second "here" and specifying the model version CESM1.2).

Line 64/65: "Other applications have not been tested so far and will be left to explore in future studies". Now that you have included this to address the reviewers comments, e.g., on more detailed analysis of the water balance but now this statement, especially mentioning "other applications"" is way to vague. Can you indeed indicate how the ExtremeX experiment results can be diagnosed in a different manner to further unravel the role of land surface processes versus circulation (and the oceans)?

This statement refers to more detailed analysis on drivers and model biases but also to the application of the experiments to different types of events. One possible application is shown in Luo et al. (2021) using the ExtremeX experiments to study biases of anomalies in near-surface temperature, precipitation, mean sea level pressure and meridional wind speed at 250hPa during certain Rossby wave events in the Northern Hemisphere.
As we explained to Reviewer #2 and state in the conclusions, the experiment setup is not ideal to study the role of the ocean but instead simulations with interactive ocean would be more suitable if the focus was on the ocean. Also, some of the drawbacks of the setup are already mentioned in the conclusions (e.g. that the analysis is co-temporal and thus a separation of initial source vs. response is not possible). If by detailed analysis on the water balance you are referring to the comment by Reviewer #1 (the first general comment), then this is something that is not possible with the present setup. We would prefer to not mix up things that might be possible with ExtremeX with analysis that would require other experiments. Reviewer #2 had the suggestion to look into the moisture budget, this is indeed one of the "other applications" we think could be mentioned here. We rephrased the lines to better explain what we mean:

"The ExtremeX experiments could also be used to examine other types of events than heatwaves. They are suitable for more in-depth analysis of model biases by examining for example the atmospheric moisture and heat budgets or the surface energy balance. In Luo et al. (2021) the ExtremeX experiments are used to study the origin of model biases in the anomaly of upper-level winds and near-surface climatology during certain summertime Rossby wave events in the Northern Hemisphere by constructing composites. Other applications have not been tested so far and will be left to explore in future studies."

Reference:
Luo, F., Selten, F., Wehrli, K., Kornhuber, K., Le Sager, P., May, W., Reerink, T. Seneviratne, S. I., Shiogama, H., Tokuda, D., Kim, H., and Coumou, D. (2021), Summertime

circumglobal Rossby waves in climate models: Small biases in upper-level circulation create substantial biases in surface imprint, *Weather and Climate Dynamics Discussions*, 1–30, doi: 10.5194/wcd-2021-48, *in review*

Line 82: for consistency use the term "observation-based data"

done

Line 85: "All experiments prescribe SSTs, sea ice and vegetation.." would be good indicate here what sea-ice and vegetation characteristics are prescribed, cover? Or other properties?

Sea ice cover fraction is prescribed and the land use (vegetation type or plant functional type in CESM) is prescribed. The sentence now reads:

"All experiments prescribe SSTs, sea ice cover fraction, and land use (i.e. vegetation) but differ in the simulation of the atmospheric circulation and soil moisture that are either interactive or constrained."

Line 92-93; had to read this sentence multiple times: suggest to change to "...experiments both components are constrained prescribing soil moisture varying over time or prescribing soil moisture using a climatological soil moisture, respectively"

We agree that the original sentence is hard to read and changed the lines based on the suggestion to say:

"... both components are constrained prescribing soil moisture time-varying or prescribing soil moisture using climatological soil moisture (but including the seasonal cycle), respectively."

Line 95: "As ExtremeX was initiated in 2017, no longer time horizon was proposed". Should have commented on this earlier but what is meant with this? I don't get what you exactly want to express with this statement.

This was solely to explain why the experiments end in 2015/2016. But as it seems to cause confusion the sentence was removed.

Line 103: "observation-based products" or "observation-based data"? be consistent

We decided to replace "observation-based products" and "observation-based values" by "observation-based data" throughout the manuscript to be consistent.

Line 113: "..to generally refer to the applied method of nudging the atmospheric large-scale circulation and prescribing soil moisture"

We agree with the suggestion made and adapted the sentence accordingly.

Line 130: "2.2.2 Prescribing Soil moisture"

We adapted the section title according to the suggestion.

Line 146: having referred before to the specific method of prescribing soil moisture and then introducing here in a very general way "2.3 Reference data sets", I would put soil moisture here under the same heading. But here you are introducing more the "Atmospheric circulation and hydrology data"

Indeed, the subchapter on soil moisture prescription is a lot about how to generate the soil moisture data that is prescribed because it is an essential part of the method description. Whereas on the other hand for the chapter on atmospheric nudging, it is more the specifications (like the relaxation time scale and the nudging profile) that have to be provided to ensure reproducibility. In the "2.3 Reference data sets" chapter we introduce the data sets that were used to validate the results of the experiments. It is true that these data sets provide information on the atmospheric circulation and hydrology. We hope that this helps to understand our choice of the structure of the manuscript and we are open for suggestions to rename and/or rearrange chapters.

Then reading the introduction again of the three models that have been included in these ExtremeX experiments, I realized that I miss(ed) a motivation on the actual of these three models. What has been the main reason that these experiments include these three different modelling systems, is there a specific reason why these models have been selected and what would be the expected differences based on the main (different) features of these models? It would be good to shortly include this in the introduction of the paper.

In the beginning we (ETH; CESM) reached out to different modeling groups that might be interested to contribute to ExtremeX. The motivation to carry out a multi-model experiment was to find out whether the results would agree. If certain phenomena/behaviour is found across all models, despite their differences, this adds to the confidence we have in the results. In the end two other modeling groups were found that were willing to run all the required experiments. Therefore, the selection of the models was in the first place due to availability of experiments and less an intentional choice of the specific models (or search for exactly these models). However, we were keen to have models that do not too closely resemble (like for example two setups of CESM). Comparing the studies by Knutti et al. (2013; for CMIP5 models) and Brunner et al. (2020; for CMIP6 models) shows that this is true for the models chosen.

Since EC-Earth3 is the most recent and was used for CMIP6, it has the highest horizontal and vertical resolution of the tree models used here and the most recent forcing data. A recent study by Brands (2022) shows that EC-Earth3 is among the best-performing models with regard to the representation of atmospheric dynamics/circulation in the Northern Hemisphere mid- to high-latitudes. MIROC5 is among the models with higher circulation errors, while CESM (comparable to the TaiESM1.0 in the paper by Brands (2022)) is in-between (for EC-Earth3 and MIROC5 this is also confirmed in the study by Fernandez-Granja et al. (2021)). Apart from that there are no fundamental structural differences between the models. Different behaviour of the models is discussed throughout the paper, we currently do not see a way to bring this to the introduction without explaining the results. The models behave differently in the CMIP experiments but as the experiments only cover the historical time period, we do not see how this might be indicative of the models' behaviour in the ExtremeX experiments.

In the introduction we added the following sentences:

"Here, building further upon the studies with CESM1.2, we present the results of the same experiments but carried out for three ESMs that were contributed each by one of the collaborating modeling groups. The models do not show high interdependence and thus are an optimal selection for a small ensemble (Brunner et al., 2020; Knutti et al., 2013). EC-

Earth3 is the most recent of the three models and, hence, has the highest horizontal and vertical resolution of the three models used here. Since EC-Earth3 was used for CMIP6, it also has the most recent forcing data. Among CMIP5 and CMIP6 models, EC-Earth3 is one of the best-performing models with regard to the representation of atmospheric circulation in the Northern Hemisphere mid-to-high latitudes (Brands, 2022; Fernandez-Granja et al., 2021). MIROC5 has contributed to CMIP5 as well as the 1.5°C versus 2.0°C global warming experiments (e.g. Hirsch et al., 2018; Mitchell et al., 2017; Shiogama et al., 2019)."

References:
Knutti, R., Masson, D., and Gettelman, A. (2013), Climate model genealogy: Generation CMIP5 and how we got there, *Geophys. Res. Lett.*, 40, 1194–1199, doi:10.1002/grl.50256.

Brunner, L., Pendergrass, A. G., Lehner, F., Merrifield, A. L., Lorenz, R. and Knutti, R. (2020), Reduced global warming from CMIP6 projections when weighting models by performance and independence, *Earth Syst. Dynam.*, 11 (4), 995–1012, doi: 10.5194/esd-11-995-2020

Brands, S. (2020), A circulation-based performance atlas of the CMIP5 and 6 models for regional climate studies in the Northern Hemisphere mid-to-high latitudes, *Geoscientific Model Development*, 15 (4), 1375–1411, doi: 10.5194/gmd-15-1375-2022

Fernandez-Granja, J.A., Casanueva, A., Bedia, J. *et al.* Improved atmospheric circulation over Europe by the new generation of CMIP6 earth system models. *Clim Dyn* **56,** 3527–3540 (2021). doi: 10.1007/s00382-021-05652-9

Hirsch, A. L., Guillod, B. P., Seneviratne,S. I., Beyerle, U., Boysen, L. R., Brovkin, V.,Davin, E. L., Doelman, J. C., Kim, H.,Mitchell, D. M., Nitta, T., Shiogama, H.,Sparrow, S., Stehfest, E., van Vuuren, D.P., & Wilson, S. (2018). Biogeophysical Impacts of Land-Use Change on Climate Extremes in Low-Emission Scenarios: Results From HAPPI-Land, Earth's Future, 6. doi: 10.1002/2017EF000744

Shiogama, H, T Hasegawa, S Fujimori, D Murakami, K Takahashi, K Tanaka, S Emori, I Kubota, M Abe, Y Imada, M Watanabe, D Mitchell, N Schaller, J Sillmann, E Fischer, J. F. Scinocca, I. Bethke, L Lierhammer Jun'ya Takakura, Tim Trautmann, Petra Döll, Sebastian Ostberg, Hannes Müller Schmied, Fahad Saeed, Carl-Friedrich Schleussner (2019) Limiting global warming to 1.5ºC will lower increases in inequalities of four hazard indicators of climate change. Environ. Res. Lett. 14, 124022. doi: 10.1088/1748-9326/ab5256

Line 233: "...some variability in wind speed between the five members.."

We changed the line to say "... some variability in wind speed and wind direction between the five members …" in order to stay consistent with Fig. A1 which shows wind speed and direction (and not u and v vectors)

Line 234: "..representation of the reference wind speed", to stress that this evaluation/validation step is mainly relying on comparison of wind speeds and not other meteo. properties

Again, this was changed to say "... representation of the reference wind speed and wind direction …"

Lines 270/271: "This contradicts the initial intuitive assumption and suggests that no sole component of the model is responsible for the biases, and hence the latter cannot be corrected in isolation". I was somewhat puzzled by what you want to express by those last three words; I guess would like to express that the biased cannot be corrected by overcoming the causes of those simulated biases by improving the model representation of only one (in isolation) of the model components". If this is indeed what you mean, possibly rephrase (more explicitly)

I believe the guess is correct. What we meant is that the biases (that we just discussed in the section; near-surface temperature and total precipitation) cannot be (substantially) corrected by improving the representation of single model components in isolation. We rephrased the sentence and for better readability we separated the statement into two sentences:

"This contradicts the initial intuitive assumption and suggests that no sole component of the model is responsible for the biases. Hence, the climatological biases that are discussed here cannot be corrected by improving the representation of the model components in isolation."

Line 292: "Nonetheless, for EC-Earth AF_SF shows larger biases than AI_SI as the temperature biases introduced in AF_SF outweigh the biases corrected, and for precipitation, the biases remain of similar magnitude". What do you mean here with biases corrected?? Corrected for what?

As AF_SF is the experiment with nudged atmosphere and prescribed soil moisture, the circulation and soil moisture should be "correct" or very close to observations. Therefore, the assumption would be that near-surface temperature and total precipitation are also well represented or at least substantially better in the AF_SF simulations compared to the AI_SI simulations (thus "corrected" with respect to AI_SI). We have changed the end of this paragraph to clarify:

"Nonetheless, for EC-Earth AF_SF shows larger climatological temperature biases than AI_SI (Fig. 4), and for precipitation, the biases remain of similar magnitude (Fig. 5). This indicates that even if some part of the temperature and precipitation biases is reduced by constraining the atmospheric circulation and soil moisture using observation-based data, other biases can be enhanced or change sign, resulting in a worse overall performance of the model."

Line 359: "which both found a weak role of the ocean in ....??" explaining the Russian heatwave or heatwaves in general?

Both studies examined the Russian heatwave. We changed the line to say:

"This result is supported by the studies by Dole et al. (2011) using initialized forecasts and Hauser et al. (2016) using an ESM, which both found a weak role of the ocean in explaining the Russian heatwave of 2010."

Line 374: "by Duchez et al. (2016) finding that the SST patterns set important preconditions for the 2015 summer", same comment here; just for the 2015 summer in general or in explaining the 2015 European summer heatwave?

Yes, again more specifically the heatwave of the 2015 summer in Europe was meant. We clarified this in the sentence:

"This is in contrast to the modeling study by Dong et al. (2016) and the observation-based study by Duchez et al. (2016) finding that the SST patterns set important preconditions for the 2015 European heatwave."

Line 399: "This is in line with the La Niña conditions that established in 2010 and 2011 and transitioned to cool to neutral during...", this statement does not read well and suggest to change to "This is in line with the La Niña conditions that prevailed in 2010 and 2011.." but want do you want to express with "transitioned to cool to neutral"?? a change to a more cool or neutrale state/contributiob bu La Nina to conditions in Australia?

Based on the Oceanic Niño Index (ONI) by the NOAA Climate Prediction Center La Niña lasted from about May-June-July 2010 to March-April-May 2012 (three month running means). Then ENSO remained neutral for the rest of 2012 and during the year 2013. The sign of the ONI was slightly negative during 2013 (but not strong enough for La Niña, this is what we wanted to express with "cool to neutral"). The sentence was altered to say:

"This is in line with the La Niña conditions that prevailed from mid-2010 to early 2012 and then remained neutral for the rest of 2012 and during 2013 (NOAA Climate Prediction Center, 2022) as well as with the findings by Lewis and Karoly (2013)."

Line 407: "whereas for the other two models the individual ratios are balanced to slightly circulation-dominated", another puzzling statement, I guess you want to express that "whereas for the other two models the individual ratios reflect that there might an even-to a slightly circulation-dominated contribution by the two main drivers of the Australian heatwave according to the experiments"

The individual ratios show about equal contributions from circulation and soil moisture for approach A and a slightly larger contribution from atmospheric circulation for approach B. We rephrased the sentence based on the suggestion to say:

"... whereas for the other two models the individual ratios reflect that the contribution by the two main drivers of the Australian heatwave is equal to slightly circulation-dominated according to the experiments (Fig. A8d)"

Line 418: "– thus not favoring a heatwave –" a would rather say "– thus not strongly enhancing the heatwave–"

We changed the sentence to say "For the 2012/2013 heatwave in Australia, all models agree that the role of the ocean is negative – thus not enhancing the heatwave – …" because "not *strongly* enhancing" reads a bit like it might in fact still be slightly enhancing the heatwave.

Line 442: "This shows the growing relative importance of the **land surface-atmosphere coupling** for long-duration events" (instead of only the state of the lanf surface, correct?)

We thank the editor for this suggestion as it indeed fits much better. The change was adopted as suggested.

Line 457: "Five experiments with varying levels of constraining were run with all models"; this is bad revised sentence, alternative; "Five experiments, with different sources of information to constrain these experiments, were run with all models"

What we meant to say was that some of the experiments are less constrained (AI_SI: only the ocean is prescribed) and some experiments are more constrained (AF_SF: ocean and soil moisture is prescribed, atmospheric circulation is nudged). We changed the existing sentence and added one sentence as follows:

"Five experiments were run with all models with one or more of the models' components being constrained. SSTs and sea ice coverage fractions are prescribed in all experiments."

Line 463: "even larger biases appear in the constrained experiments", alternative also given that there were already biases in the default experiments, "the biases are even further enhanced in some of the constrained experiments"

In most cases it does not look like already existing biases are being enhanced in the constrained experiments. Instead, the magnitude of the bias is usually the same or slightly reduced (compare for example the temperature bias in Fig. 4a between AI_SI and AF_SI; for all three models the sign of the bias is the same in the two experiments and the RMSE is the same or slightly smaller in AF_SI). There are regions where the constrained experiments show biases of the opposite sign (see for example the positive precipitation bias in Europe and western Asia from 30°N to 60°N for AF_SI from MIROC in Fig. 5a or the negative precipitation bias east of the Black Sea for AI_SF from EC-Earth in Fig. 4a). Nevertheless, enhancement of already existing biases also happens, for example, negative temperature biases north of ~60°N seem to be enhanced in AI_SF in EC-Earth compared to AI_SI. To better address all of these points we changed the sentence to say:

"In some cases, biases are enhanced or even change sign in the constrained experiments (Fig. 3, Fig. 4 and Fig. 5)"

Line 468: "including atmospheric circulation and soil moisture (dynamics) interactions"

The suggested change works fine for us and was adopted in the manuscript.

Line 474: "For events that are mainly ocean-driven, we would recommend a setup with interactive ocean experiments to compute the ocean contribution more accurately." Could you possibly shortly indicate/ hypothesize on extreme events that might strongly affected by ocean-atmosphere-driven process interactions. Would there be a role in heatwaves (in suppressing, or, enhancing) or other extreme events?

We think that this can affect heatwaves but also other types of extreme events such as floods or droughts. Essentially, all events strongly driven by ENSO, or other coupled ocean-atmosphere phenomena such as the Indian Ocean Dipole or the Pacific Decadal Oscillation would benefit from a better estimation of the ocean contribution by using interactive ocean experiments. We added the following sentence to the (new) Line 496 of the revised manuscript:

"This would apply for example to extreme events (i.e. droughts, heatwaves, floods) that are strongly driven by the El Niño Southern Oscillation and other coupled ocean-atmosphere phenomena such as the Indian Ocean Dipole or the Pacific Decadal Oscillation."

Line 476: "..the potential ocean contribution.."

done

Line 493: "The largest contribution **by recent warming** is found for the U.S. heatwave.."

Yes, thank you, the sentence was changed accordingly.